# Global distribution, trends, and drivers of flash drought occurrence

Jordan I. Christian [1✉], Jeffrey B. Basara [1,2], Eric D. Hunt[3], Jason A. Otkin [4], Jason C. Furtado[1], Vimal Mishra [5,6], Xiangming Xiao [7] & Robb M. Randall [8]

Flash drought is characterized by a period of rapid drought intensification with impacts on agriculture, water resources, ecosystems, and the human environment. Addressing these challenges requires a fundamental understanding of flash drought occurrence. This study identifies global hotspots for flash drought from 1980–2015 via anomalies in evaporative stress and the standardized evaporative stress ratio. Flash drought hotspots exist over Brazil, the Sahel, the Great Rift Valley, and India, with notable local hotspots over the central United States, southwestern Russia, and northeastern China. Six of the fifteen study regions experienced a statistically significant increase in flash drought during 1980–2015. In contrast, three study regions witnessed a significant decline in flash drought frequency. Finally, the results illustrate that multiple pathways of research are needed to further our understanding of the regional drivers of flash drought and the complex interactions between flash drought and socioeconomic impacts.

[1] School of Meteorology, University of Oklahoma, Norman, OK, USA. [2] School of Civil Engineering and Environmental Science, University of Oklahoma, Norman, OK, USA. [3] Atmospheric and Environmental Research, Inc., Lexington, MA, USA. [4] Cooperative Institute for Meteorological Satellite Studies, Space Science and Engineering Center, University of Wisconsin-Madison, Madison, WI, USA. [5] Civil Engineering, Indian Institute of Technology (IIT), Gandhinagar, India. [6] Earth Sciences, Indian Institute of Technology (IIT), Gandhinagar, India. [7] Department of Microbiology and Plant Biology, Center of Spatial Analysis, University of Oklahoma, Norman, OK, USA. [8] CCDC Army Research Laboratory, White Sands Missile Range, Mexico City, NM, USA. ✉email: jchristian@ou.edu

Flash drought is a critical sub-seasonal phenomenon that exhibits multifaceted challenges to agriculture, the economy, and society[1]. Given the rapid land-surface desiccation associated with flash drought, the agricultural sector can be devastated and experience substantial economic damage due to lower crop yields and curtailed livestock production[1–4]. Rapid drought intensification can severely impact ecosystems via excessive evaporative stress on the environment[5–10] and contribute to compound extreme events with cascading impacts including an increased risk for wildfire development, depletion of water resources, reduction of air quality, and decreased food security[11–16].

With a wide range of impacts associated with flash drought and challenges related to its sub-seasonal prediction[17,18], a critical goal within the scientific community is to advance knowledge of flash drought events. As such, research has been undertaken to improve the detection, evaluation, and monitoring of flash drought, including sub-surface analysis with soil moisture[19], atmospheric evaporative demand[8,20], evaporative stress via evapotranspiration (ET) and potential evapotranspiration (PET[6,7,9]), and impact-based approaches[21]. In addition, rapid drought intensification has been identified across the United States[3,6,22], Brazil[23], southern Africa[24], Spain[25], western Russia[15], and Australia[10]. A critical next step that builds upon these regional studies is to quantify the global distribution of flash drought, the seasonal frequency of flash drought, the trends in the occurrence of rapid intensification toward drought, and the drivers of flash drought development.

While recent progress in flash drought research has been accomplished via case studies and regional analyses, a key scientific question remains: What global regions are the most susceptible to flash drought occurrence? To address this question, the spatial distribution of flash drought events was identified via four global reanalysis data sets for the period spanning 1980–2015. The results presented here reveal[1] the regions with the strongest, reanalysis-based consensus for hotspots of flash drought development[2], the seasonal characteristics of flash drought frequency[3], the trends in flash drought spatial coverage, and[4] the relative drivers of flash drought occurrence. Following the results, the implications of global flash drought hotpots are discussed, including the possible physical mechanisms that drive rapid drought intensification and the societal impacts associated with rapid drought intensification.

## Results

**Global flash drought occurrence.** Evapotranspiration (ET) and potential evapotranspiration (PET) were used from four reanalysis data sets (Modern-Era Retrospective analysis for Research and Applications: MERRA[26]; MERRA, Version 2: MERRA-2[27]; ERA-Interim[28]; ERA5[29]) to quantify the standardized evaporative stress ratio (SESR; the ratio between evapotranspiration and potential evapotranspiration[9]). SESR represents the overall evaporative stress on the environment. SESR becomes positive when ample soil moisture is available, surface temperatures and vapor pressure deficit are lower, and cloudy skies are present (reduced shortwave radiation). In contrast, SESR becomes negative when soil moisture is depleted, surface temperatures and vapor pressure deficit increase, and clear skies are present (increased shortwave radiation). SESR is similar to the evaporative stress index (ESI[30,31]) in which both indices are calculated by the ratio of ET and PET and then standardized. However, SESR is primarily derived using reanalysis-based variables while the ESI is derived using satellite observations. After SESR was calculated, SESR was processed through a comprehensive flash drought identification methodology that incorporates multiple criteria associated with

rapid intensification toward drought (the flash component of flash drought) and impact (the drought component of flash drought[9]). As a methodology for evaluating flash drought, SESR compares well with the satellite-based ESI[9,30,31], acts as an early drought indicator, and corresponds with impacts indicated by the United States Drought Monitor (USDM[9,32]) and land-surface desiccation via satellite observations[15]. Further, it provides flash drought occurrence both regionally and nationally across the United States[9,33] and represents the development and evolution of flash drought case studies using different data sets across different regions around the globe[3,15].

The regions with the highest frequency of flash drought occurrence were primarily found within the tropics and subtropics (Fig. 1). These locations include a large portion of Brazil, the Sahel, the Great Rift Valley, and India, with composite flash drought occurrence between 30 and 40% of the years within the 36-year time period (1980–2015) of analysis. Three of these four major hotspots for flash drought occurrence had coefficients of variation below 0.3 throughout most of their domains (the Sahel, the Great Rift Valley, and India), indicating strong agreement between the four reanalysis data sets (Supplementary Fig. 1). Additional areas within the tropics that had lesser, but notable flash drought occurrence included central Mexico, the Indochinese Peninsula, and northern Australia, with flash drought occurrence between 20 and 30% of the years. For these regions, the Indochinese Peninsula and northern Australia had strong agreement between data sets (coefficients of variation <0.3; Supplementary Fig. 1). In the mid-latitudes, local hotspots of flash drought occurrence (10–20%) exist across the central United States, Iberian Peninsula, Asia Minor, southwestern Russia, and northeastern China. These regions exhibited larger variability between reanalyses (coefficients of variations between 0.3 and 0.6), with notable disagreement in flash drought occurrence across the central United States (Supplementary Fig. 1).

**Temporal flash drought characteristics.** The onset and timing of flash drought is a critical component to agricultural impacts, as flash drought can drastically reduce crop yields and lead to severe economic losses, and potentially disrupt food security. As such, the monthly distribution of flash drought occurrence was examined across 1) global flash drought hotspots and/or 2) regions with extensive crop cultivation (Fig. 2). Study regions were selected over global hotspots where flash drought occurred in more than 30% of the study years. These regions included Brazil, the Sahel, the Great Rift Valley, and India. Additional study regions were examined where a regional maximum in flash drought frequency exceeded 15%, including the central United States, central Mexico, the Iberian Peninsula, Asia Minor, southwestern Russia, the Indochinese Peninsula, northeastern China, and northern Australia. Three additional regions were investigated to quantify flash drought seasonality over tropical rainforest (the Amazon) and the Southern Hemisphere mid-latitudes (Argentina and southeastern Australia). Of the 15 locations analyzed, eight were both a regional maximum in flash drought occurrence and regions of major agricultural production with croplands covering at least 20% of the total land area in a given domain (Fig. 1). These regions included the Corn Belt across the Midwestern United States, barley production in the Iberian Peninsula, the western Russian wheat belt, wheat production in Asia Minor, rice-producing regions in India and the Indochinese Peninsula, maize production in northeastern China, and millet and sorghum production across the Sahel[34]. Two additional areas did not necessarily exhibit a local hotspot in flash drought occurrence but are significant agricultural locations (Fig. 1a). These regions include maize and wheat production

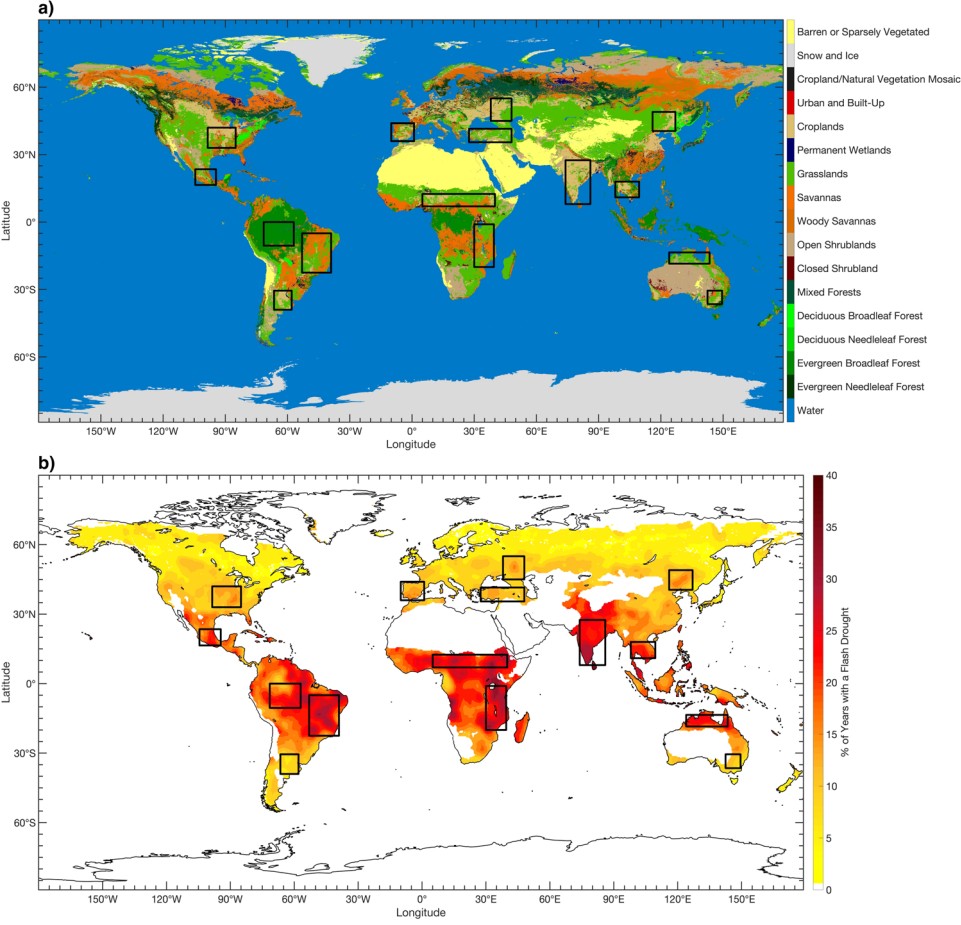

**Fig. 1 Land cover type and flash drought occurrence. a** Land cover type (MCD12C1) is given by MODIS version 6 for 2015 using the International Geosphere-Biosphere Programme classification. **b** Mean flash drought frequency from the four reanalysis data sets is represented as the percent of years with a flash drought between 1980 and 2015. The black outlines represent domains used for the temporal analysis in Figs. 2–4.

across the Pampas in Argentina and wheat production in southeastern Australia[34].

For each of the regions, flash drought events were partitioned by the month in which rapid drought development began. The frequency for each month was calculated as the percent of each month's contribution to the annual total of flash drought occurrence for that region. For the regions located in the tropics and subtropics (between 30°S and 30°N) flash drought occurrence was examined year-round, while regions in the mid-latitudes (between 30 and 60°) were investigated for flash drought occurrence within their approximate growing season (i.e., March through October in the Northern Hemisphere and September through April in the Southern Hemisphere).

For a majority of regions within the mid-latitudes in the Northern Hemisphere, a seasonality in flash drought frequency is evident within the growing season with flash droughts most likely between May and July for the central United States, southwestern Russia, and northeastern China (Fig. 2). An exception to this seasonality occurs across the Iberian Peninsula and Asia Minor. A bimodal distribution of flash drought occurrence is seen for the Iberian Peninsula with peaks in flash drought frequency in June and September, while the occurrence of flash drought in Asia Minor generally increases throughout the growing season. For the Southern Hemisphere mid-latitude regions, the monthly distribution of flash drought frequency differs from the primary seasonal pattern seen in the Northern Hemisphere mid-latitudes (a peak in flash drought frequency in the late spring and early summer). For example, agricultural regions in Argentina display

monthly variability in flash drought occurrence, while southeastern Australia exhibits a peak in flash drought occurrence near the end of the austral growing season (Fig. 2).

A seasonality in flash drought frequency is also evident in regions located within the tropics and subtropics, with the phase of their pattern dependent upon the hemisphere in which they reside. For example, the four regions in the Northern Hemisphere tropics and subtropics (Mexico, the Sahel, India, and the Indochinese Peninsula) generally had their highest occurrence of flash drought in the boreal growing season (Fig. 2). Three of the four regions in the Southern Hemisphere tropics (Brazil, the Great Rift Valley, and northern Australia) exhibit peak flash drought occurrence during the austral growing season.

**Changes in spatial coverage of flash drought**. To quantify the change in flash drought coverage with time, flash drought spatial coverage was calculated for each domain and year. The Mann–Kendall test was applied to each time series to determine if statistically significant trends were evident in yearly flash drought coverage. Of the 15 regions investigated from the composite analysis in this study, six regions (the central United States, Iberian Peninsula, Asia Minor, Brazil, the Sahel, and southeastern Australia) had a statistically significant ($p < 0.1$) increasing trend in flash drought coverage, while three regions (India, the Great Rift Valley, and northern Australia) had a statistically significant ($p < 0.1$) decreasing trend (Fig. 3). In addition, four of the nine regions identified as having statistically significant trends associated with the composite analysis had at least three individual

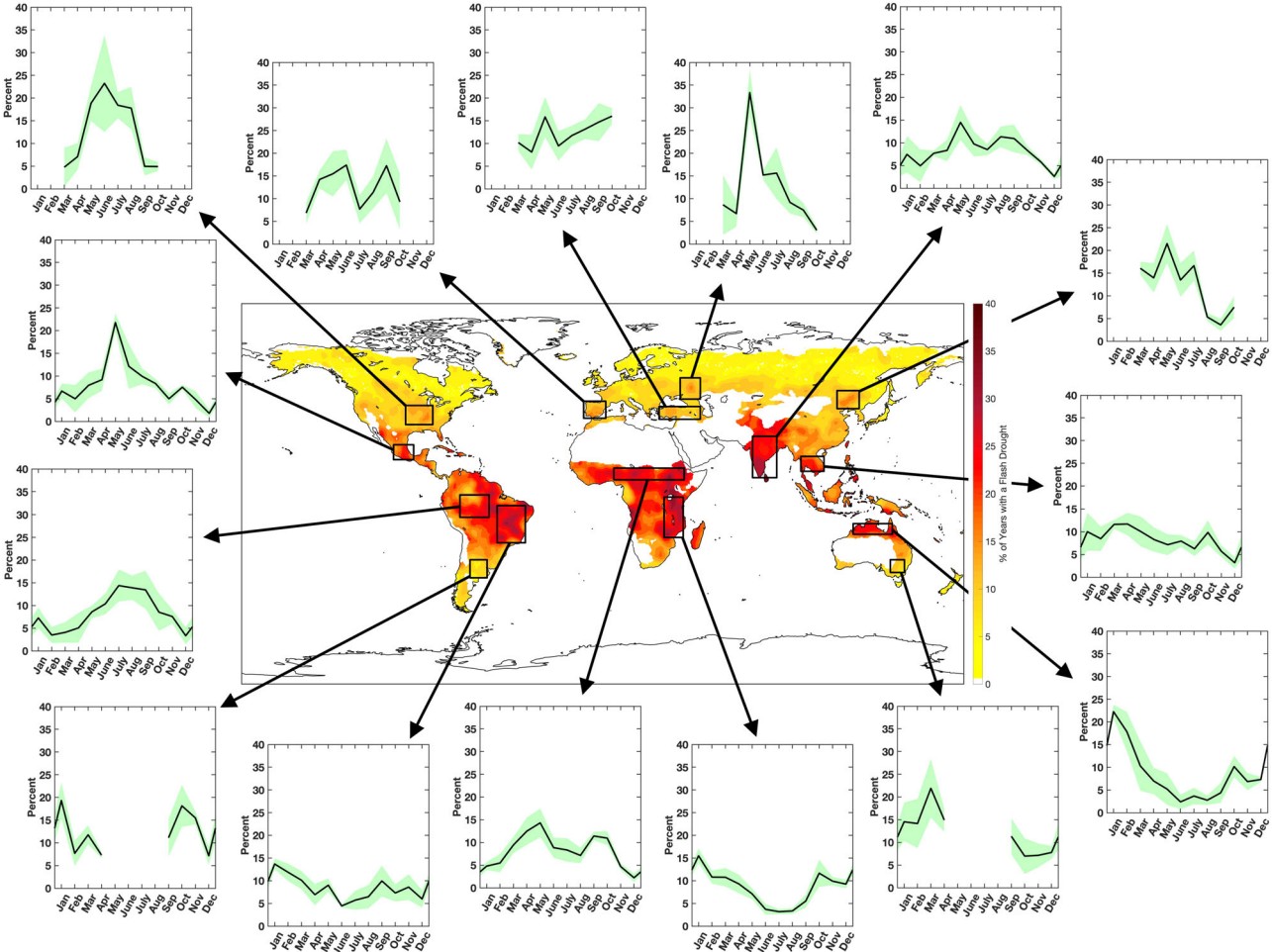

**Fig. 2 Monthly flash drought occurrence.** Monthly distribution of mean flash drought occurrence from the four reanalysis data sets (black line) for each domain outlined in black on the map. The green shaded area represents the variability (standard deviation) between the four reanalyses.

reanalysis data sets produce statistically significant trends (Supplementary Fig. 2). Overall, six of the nine regions had all four individual reanalyses indicate consistent directions of the trend (positive or negative), while two additional regions had three of the four reanalyses with consistent signs of the trend.

Each of the regions also exhibited varying magnitudes of the trend. For example, the central United States and Iberian Peninsula had modest changes (~4%) in flash drought spatial coverage associated with statistically significant trends during the study period (Fig. 3). In contrast, Asia Minor, India, the Sahel, the Great Rift Valley, and northern Australia had large changes in spatial coverage, with changes during the 36-year period between 14 and 26%. A few regions also had minimal changes in flash drought spatial coverage with time, including Mexico, Argentina, and the Indochinese peninsula. Each of these regions had spatial coverage changes <2% over the 36-year period. The overall change from the composite analysis for the Amazon revealed a decreasing trend that was not statically significant. However, large spatial areas of flash drought coverage prior to 2000 in the MERRA data set produced large variability between the data sets.

It is important to note that the results of the trend analysis only apply to the 36-year period used in the study (1980–2015) and do not indicate that these trends will extend into the future. Further, notable trends revealed for the analysis may also be embedded within the internal variability of the climate due to the relatively short study period and may change with a longer period of record.

**Drivers of flash drought development.** Rapid drought intensification occurs via two key drivers: a critical lack of precipitation and increased evaporative demand[1]. When a precipitation deficit occurs over an extended period of time (e.g., several weeks), soil moisture is depleted by evapotranspiration yielding increased evaporative stress and the potential for desiccation of the terrestrial surface. In addition, persistent atmospheric conditions can amplify evaporative demand at the land surface via increased solar insolation and vapor pressure deficit thereby increasing the PET and evaporative stress. While the combination of precipitation deficits and positive PET anomalies are well known to promote flash drought development, the relative contribution of each driver toward rapid drought intensification is relatively unknown.

To quantify the contribution of large precipitation and PET anomalies toward flash drought development, the standardized precipitation index (SPI[35,36]) and standardized PET anomalies were calculated during the time frame of each individual flash drought event within the 15 study regions (Fig. 4). The frequency in which large anomalies occurred during flash drought was determined for (a) SPI values less than or equal to −1, (b) standardized PET anomalies greater than or equal to 1, (c) both an SPI anomaly and PET anomalies that occurred concurrently during rapid drought development, and (d) at least one anomaly occurring during the flash drought.

A key result from the global analysis is that on average across the 15 study regions, large, positive evaporative demand

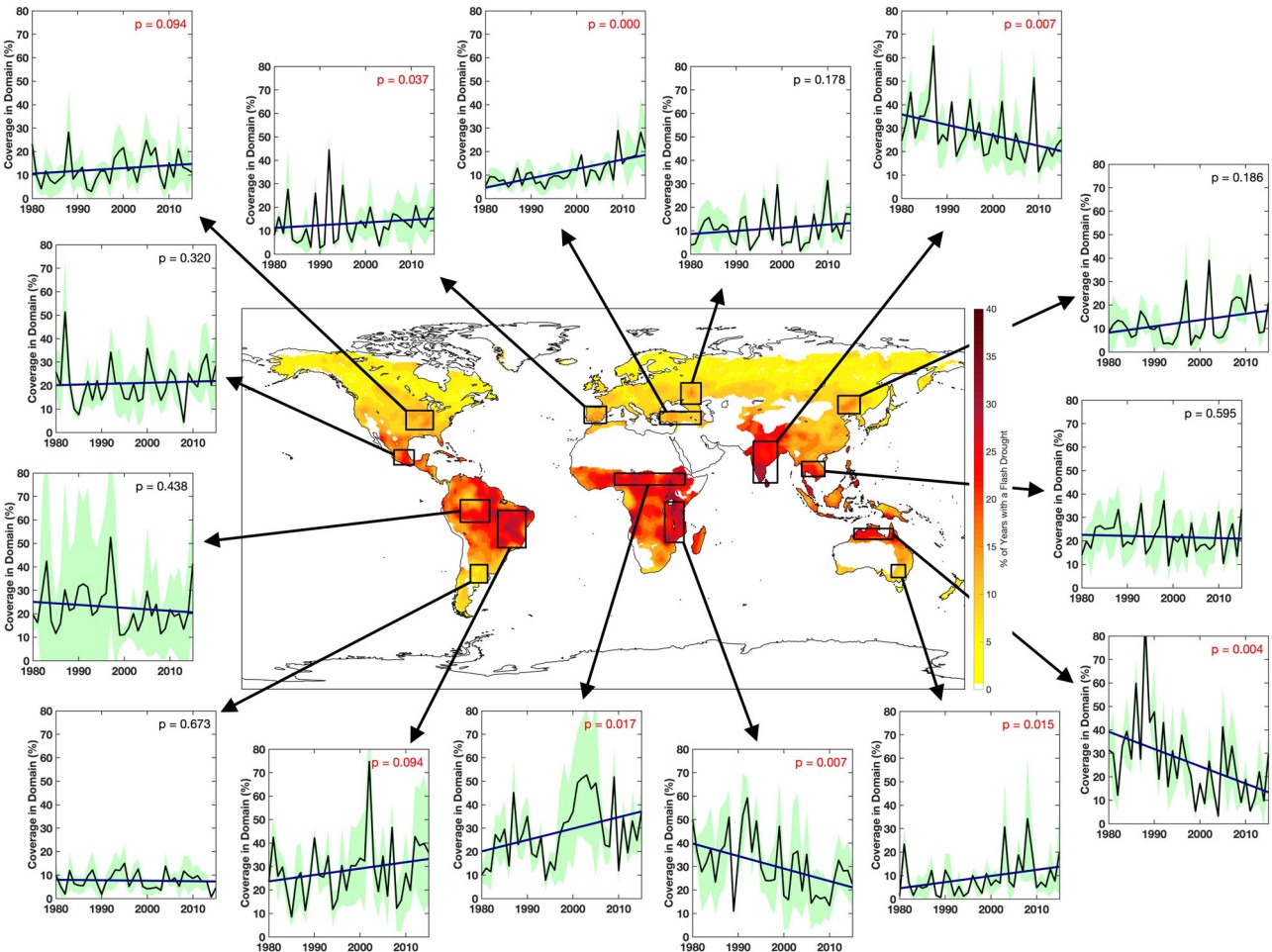

**Fig. 3 Trends in flash drought occurrence.** Mean flash drought spatial coverage (percent) from the four reanalysis data sets (black line) for each of the domains outlined in black on the map. The green shaded area represents the variability (standard deviation) between the four reanalyses and the thicker blue line represents the trend line for flash drought spatial coverage. *p*-values highlighted in red are statistically significant trends at the 90% confidence level using the Mann–Kendall test.

anomalies occurred during flash drought events at a similar rate compared to large precipitation deficits (33% of flash drought events for SPI, 31% of flash drought events for PET; Fig. 4). Further, a large SPI or PET anomaly occurred during 44% of flash drought events on average across the study regions. However, the lead driver during flash drought development varies regionally across the globe. For example, the three study regions across Europe (the Iberian Peninsula, Asia Minor, and western Russia) exhibited a higher rate of PET anomalies in flash drought events compared to large precipitation deficits (Supplementary Fig. 3). In contrast, study regions across the Americas (central United States, central Mexico, the Amazon, Brazil, and Argentina) have negative precipitation anomalies as the leading driver of flash drought.

In addition to the individual contributions of precipitation deficits and enhanced evaporative demand on flash drought development, the concurrent contribution of below-average precipitation and above-average PET was also examined for the 15 study regions. Approximately 20% of all flash drought events across the study domains exhibited both a large precipitation deficit and positive PET anomaly during flash drought development. While most of the mid-latitude study regions in the Northern and Southern Hemisphere had a relatively low frequency of SPI and PET anomalies occurring at the same time during flash drought (5–16%, excluding the central United States), domains in the tropics and subtropics had a notably higher frequency (20–34%).

## Discussion

Preferential regions for flash drought development across the globe were revealed in the climatological analysis of rapid drought development (Fig. 1). While limited studies have examined climatological flash drought occurrence, flash droughts have been identified across the Northern Hemisphere with two major hotspots (the Sahel and India[37]), which is consistent with the findings in this study (Fig. 1). In addition, a mid-latitude band of enhanced flash drought occurrence across Europe and Asia is evident in both studies. A notable difference is located over North America, where the prior study shows a global hotspot of flash drought occurrence in the southern United States and northern Mexico[37]. The results here reveal a higher frequency of flash drought occurrence in the central and Midwestern United States, with another local maximum in frequency over central Mexico (Fig. 1). A similar belt of enhanced flash drought risk across the central United States has also been shown in previous studies[9,21] (Fig. 1).

While SESR and the flash drought methodology used in this study have consistently shown to identify flash drought[3,9,15,33] for several notable events across different regions, it is critical to evaluate its performance in capturing flash drought with respect to land-surface impact. A key hydrological variable used to determine vegetative impact during flash drought analysis is soil moisture[14,19]. While SESR indirectly includes soil water content via the magnitude of ET, examining soil moisture directly

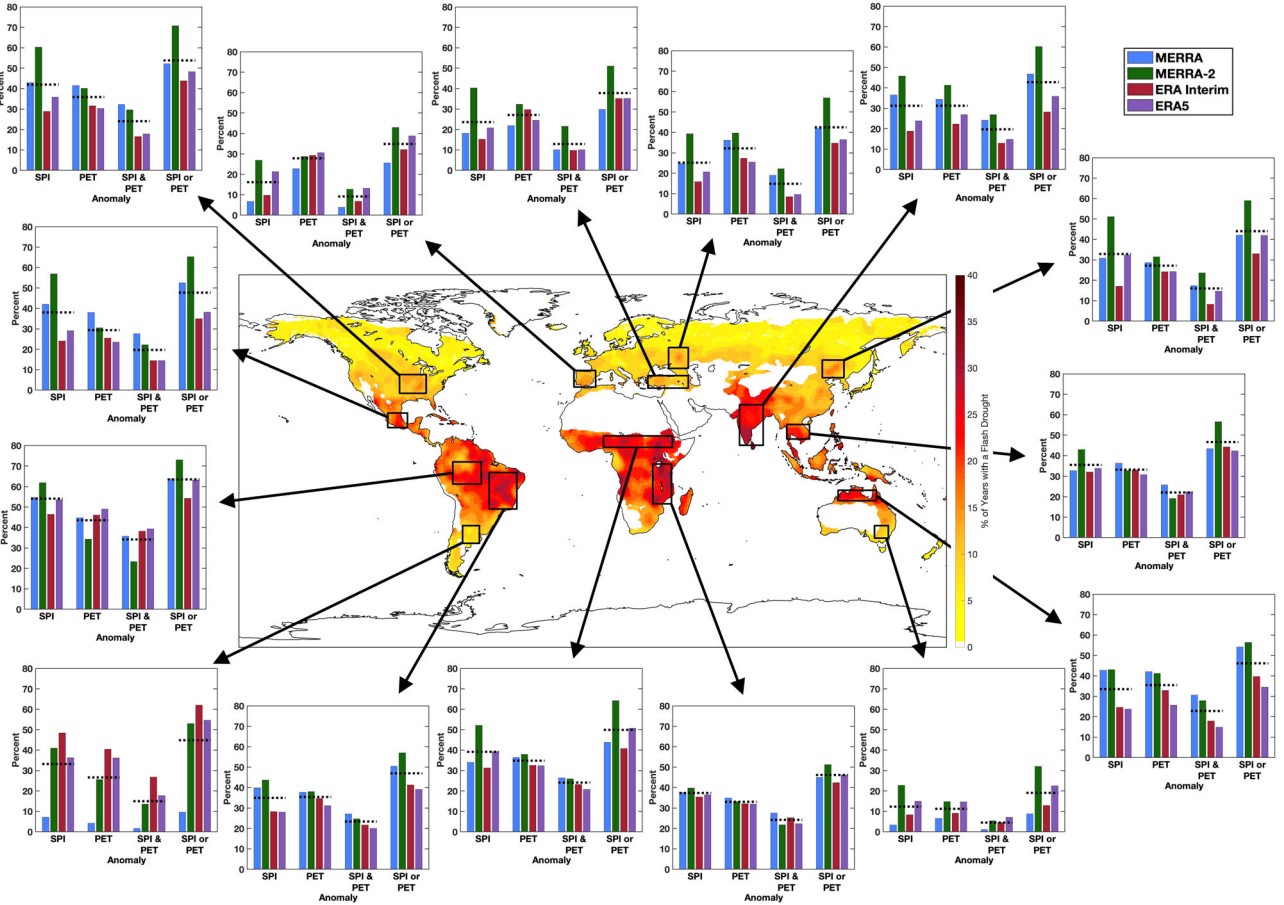

**Fig. 4 Drivers of flash drought occurrence.** Percentage of flash drought events with an SPI anomaly (SPI $\leq -1$), a PET anomaly (PET $\geq 1$), an SPI and PET anomaly (SPI $\leq -1$ and PET $\geq 1$), and an SPI or PET anomaly (SPI $\leq -1$ or PET $\geq 1$) from the four reanalysis data sets (different colored bars) for each of the domains outlined in black on the map. The black dotted lines represent the mean between all four reanalysis data sets.

provides insight into the magnitude of land-surface desiccation from SESR-derived flash drought events. After calculating the average soil moisture percentile at the end of all flash drought events between the four reanalysis data sets, it was found that soil moisture was depleted to the 20th percentile or lower in 11 of the 15 study regions and depleted to the 25th percentile or lower in some portion of all 15 study regions (Supplementary Fig. 4). Overall, the analysis indicates that flash droughts identified via evaporative stress consistently capture rapid drought intensification that leads toward depleted soil moisture content.

Flash drought development is driven by the simultaneous occurrence of precipitation deficits and above-average evaporative demand[1]. The results in Fig. 4 and Supplementary Fig. 3 show strong evidence for the nearly equal importance of highly anomalous PET and a lack of precipitation during flash drought. Enhanced PET was the lead driver during flash drought development (47.8% of the time) nearly as often as negative precipitation anomalies (52.2% of the time; Supplementary Fig. 3). As such, while large precipitation deficits remain a key driver in flash drought development, excessive evaporative demand at the land surface can also strongly contribute toward rapid drought intensification when only small to moderate precipitation deficits occur during the flash drought development period.

In regions where the contributions of large SPI and PET anomalies are low, soil type and land cover type may increase the complexity of flash drought development[9]. Of particular note are the study regions over southeastern Australia, the Iberian Peninsula, and Asia Minor which had the lowest contributions

from SPI and PET anomalies during flash drought. These three study regions are located in semi-arid environments that either directly border arid/hyper-arid environments or barren regions (Fig. 1a). As such, sparse vegetation, soils with weak water retention, and advection from arid or hyper-arid regions may complicate flash drought development in these regions with respect to rapid drought intensification primarily associated with large precipitation deficits and above-average evaporative demand.

Several factors contribute to the preferential occurrence of flash drought hotspot regions across the globe. The first of these is the role of land–atmosphere coupling in flash drought development[3,9,12,15]. While local land–atmosphere interactions are very complex[38], the fundamental relationship between flash drought development and land–atmosphere coupling can be summarized with key moisture and thermal variables. As soil moisture is depleted, ET into the atmosphere decreases. Concurrently, the effective moderation of land-surface temperatures by ET is limited, thereby further increasing evaporative demand. Reduced moisture flux from the surface contributes to a drier atmospheric column, which inhibits the generation of precipitation. This positive feedback process of drying the land surface, increasing surface temperatures, and lowering the potential for precipitation aids flash drought development. As such, many of the global hotspots for flash drought identified in this study are also located over regions with an enhanced signal of land–atmosphere coupling. These regions include the central United States, the Sahel, and India (Fig. 1[39–42]). This overlap of high flash drought occurrence and enhanced land–atmosphere coupling indicates that

land–atmosphere coupling may have a critical role in rapid drought development, especially in flash drought hotspots that lie in climate transitions zones and are sensitive to coupling dynamics.

Anticyclones are also an important contributor to flash drought development. Through subsidence and the associated suppression of rainfall, upper-level ridges can limit the potential for soil moisture replenishment. Concurrently, less cloud coverage and warmer surface temperatures increase the evaporative demand of moisture from the land surface. As such, anticyclones have a dual impact on increasing evaporative stress by limiting moisture availability for ET and increasing PET. An increased risk for flash drought development is particularly evident with blocking highs that persist for several weeks. Examples of blocking highs contributing to flash drought development include the 2012 central US flash drought and 2010 western Russian flash drought, where a lack of rainfall and increased evaporative demand associated with a blocking high set the foundation for rapid drought development[15,18].

In addition to the contributions of sub-seasonal features on flash drought development (e.g., land–atmosphere coupling and blocking highs), climatic features can also influence the spatial distribution of flash drought events revealed from the composite analysis. An example of this is associated with average daily PET across the globe. In the tropics and subtropics, the average daily PET exceeds 5 mm/day (Supplementary Fig. 5). By contrast, a majority of land areas in the mid-latitudes experience smaller daily averaged PET compared to the tropics, between 3 and 5 mm/day (Supplementary Fig. 5). Given that larger values of evaporative demand will increase the upper limit for the rate of ET, flash drought development would most likely occur in regions with consistently high PET and result in a greater potential for rapid increases in evaporative stress on the environment. As such, the overall higher frequency of flash drought hotspots in the tropics (30–40%) as compared to the mid-latitudes (15–20%) may be attributed to climatologically higher values of evaporative demand in the tropics and subtropics (Fig. 1).

Regions with relatively high interannual variability in rainfall also have a tendency for increased flash drought risk. For example, the tropics experience a higher frequency of flash drought events (e.g., equatorial South America and Africa) and higher precipitation variability compared to mid-latitudes regions (Fig. 1 and Supplementary Fig. 6b). The relationship between high precipitation variability and flash drought occurrence is also connected to the higher levels of evaporative demand seen in the tropics and subtropics (Supplementary Fig. 5). Even with high annual precipitation amounts (e.g., >175 cm per year) moderating potential flash drought development near-equatorial regions in South America and Africa (Supplementary Fig. 6a), large interannual variability in precipitation coupled with persistently high evaporative demand throughout the year provides strong potential for flash drought development in these regions.

As discussed, a diverse set of meteorological and climatic drivers contribute to preferential regions for flash drought development. Similarly, various drivers will also contribute to the seasonality of flash drought occurrence. For example, the Asian-Australian monsoon provides extensive precipitation across India, eastern/southeast Asia, and northern Australia. The Asian monsoon typically begins in June and continues throughout boreal summer, providing more than 57% of the total annual rainfall in these regions[43]. Across India, a percentile-based methodology using soil moisture was used to identify flash droughts during the monsoon and non-monsoon seasons, with the majority of the flash drought events occurring during the monsoon season, especially across the central, northwest, and northeast regions of India[44]. A similar result was found using SESR over the India domain in this study, with flash droughts

primarily initiating between May and September (Fig. 2). Likewise, northern Australia receives ~80% of its annual mean precipitation from the monsoon during austral summer (November through April[45]). From the analysis of seasonal flash drought occurrence, these regions experience their peak frequency at the beginning of their respective monsoon seasons (Fig. 2). Thus, a delay, absence, or reduction of monsoon rainfall can significantly contribute to flash drought development, provided above-normal evaporative demand is also present to promote rapid land-surface desiccation.

Another example of a flash drought driver can be examined in the Sahel, where the oscillation of the Inter-Tropical Convergence Zone (ITCZ) and the onset of the West African monsoon are the primary contributors to rainfall and intraseasonal rainfall variability across this region. The onset of monsoon and ITCZ-induced rainfall generally begins in late June across the Sahel[46,47] and the timing of this onset corresponds with a peak in flash drought risk seen across the Sahel in May (Fig. 2). This timing indicates that flash drought development is more likely to occur in the May to June transition period associated with increasing climatological rainfall, especially if the onset of ITCZ-induced and monsoon rainfall is delayed or significantly reduced, in combination with increased evaporative demand. The secondary peak of flash drought occurrence in September and October is likely related to the cessation of rainfall associated with the southward shift of the ITCZ (Fig. 2). Below-average precipitation accumulation coupled with above-average evaporative demand during a time frame that receives relatively small amounts of rainfall (e.g., ~5 cm of rainfall in September and 1–2 cm of rainfall in October on average[47]) will increase the likelihood of flash drought development during this time of the year.

A unique example of seasonality of flash drought occurrence is seen in the study region over the Amazon. Unlike some regions where monthly peaks in flash drought frequency can be attributed to intraseasonal drivers of precipitation variability (e.g., monsoons and the ITCZ), flash drought occurs most often in the dry season across the Amazon (July through September; Fig. 2). The increased frequency of flash drought can be related to vegetation dynamics and atmospheric conditions, in which vegetation with greater photosynthetic capacity and increased solar radiation due to a lack of precipitation and clouds occurs in the dry season[48]. Overall, the coupled effect of increased evaporative demand, limited rainfall, and increased ET resulting in a rapid soil moisture depletion during the Amazonian dry season may enhance the likelihood of flash drought development during this time frame.

Changes in flash drought occurrence, as a function of evaporative stress, can be examined from two perspectives: changes in ET with time or changes in evaporative demand (PET) with time. Increases in PET can be related to global climate change, with increases in surface temperature and the vapor pressure deficit being critical factors[49,50]. Locations that have increased evaporative demand will have a greater risk for flash drought development through enhanced evaporative stress. Regions that have experienced statistically significant ($p < 0.1$) increasing trends in flash drought spatial extent (Fig. 3) and statistically significant ($p < 0.1$) increasing trends in PET during the growing season (Supplementary Fig. 7) include the Iberian Peninsula, Brazil, and the Sahel. The risk for flash drought development may continue to increase in certain locations due to the effect of increased evaporative demand as increases in PET are expected in a future warming climate[51]. In contrast, locations with climatological increases in precipitation will have greater availability of soil moisture for ET, which will mitigate the enhanced evaporative stress and reduce opportunities for rapid drought intensification. Regions such as India and northern Australia may have

decreased flash drought spatial coverage over the last several decades due to changes in the magnitude and timing of precipitation (Fig. 3). For example, decreases in the South Asian monsoon circulation have contributed to changes in mean precipitation and variability during the summer across India[52], while changes in the intensity of the Walker circulation may contribute to changes in precipitation over the Maritime Continent and northern Australia[53,54]. Climate features such as these may have a critical role in reducing the likelihood of flash drought development over time.

Teleconnections can also have a significant role in the long-term (interannual and decadal) variability of flash drought spatial coverage. If a region's climate (e.g., temperature and precipitation) is sensitive to a particular teleconnection, the potential for flash drought development could change. For example, a teleconnection phase that promotes drier and warmer conditions for a specific region, especially during the growing season when evaporative demand is higher, may increase flash drought frequency/coverage for the time period within that phase. By contrast, wetter and colder conditions may decrease flash drought development. An example of this relationship is shown in Supplementary Fig. 8, where yearly flash drought spatial extent has a statistically significant ($p < 0.1$) correlation with the December–February averaged Niño 3.4 index (i.e., sea surface temperature anomalies averaged between 5°N–5°S and 120°W–170°W), a proxy for the El Niño-Southern Oscillation (ENSO), over the Amazon, Argentina, the Indochinese Peninsula, and the Great Rift Valley. In addition to teleconnections with interannual periodicities (e.g., ENSO with dominant frequencies of 2–7 years), teleconnections with interdecadal variability (e.g., the Pacific Decadal Oscillation[55]) may superpose a long-term cyclic signal on climatological flash drought occurrence. However, investigation of these signals would require a data set that is longer than that of reanalysis data from the satellite era (1979–present).

Many of the meteorological drivers and climatic features previously discussed (land–atmosphere coupling, anticyclones, interannual variability of rainfall, monsoons, the ITCZ, and ENSO) can also contribute toward conventional drought development (i.e., drought development on seasonal timescales or longer[56–58]). However, while drought is primarily characterized by a lack of precipitation, flash drought development occurs due to a combination of below-average precipitation and enhanced evaporative demand[1]. As such, the unique contribution of these features toward flash drought development involves not only the suppression of rainfall, but the additional influence of above-average evaporative demand to rapidly deplete moisture and lead to rapid land-surface desiccation.

The climatology of flash droughts provided in this study is derived from evaporative stress. While evaporative stress is related to other hydrologic variables used for flash drought analysis (e.g., soil moisture), it is important to note that the results of this study may differ from those that use a different variable or flash drought identification methodology. However, key hotspots shown in this study align with a previous study using soil moisture and a different identification methodology for the Northern Hemisphere[37], indicating the consistency of major flash drought hotspots regardless of the variable or methodology used. Local hotspot regions that vary between evaporative stress-driven flash drought and soil moisture-driven flash drought suggest a greater complexity of flash drought development in these regions. As such, the results and conclusions in this study should be primarily limited to evaporative stress-based flash drought events.

The analysis presented here reveals 1) the preferential regions for flash drought across the globe, 2) the seasonality of flash drought occurrence for selected hotspots and agricultural regions,

3) notable trends in flash drought spatial coverage for the examined locations, and 4) the contribution of key drivers in flash drought development. While flash drought frequency varies significantly across the globe, nearly every region experiences rapid drought development (excluding arid and cold regions; Fig. 1). Furthermore, above-average evaporative demand and precipitation deficits contribute with similar frequency to flash drought development (Fig. 4). Importantly, a majority of the regional hotspots of flash drought occurrence are regions with extensive agriculture production (Fig. 1). In addition to flash drought frequency, seven out of the twelve hotspot regions had statistically significant trends and are also associated with major crop production (Fig. 3).

A common theme associated with flash drought development is the impact on crop yields. Yield losses occur through rapid depletion of root zone soil moisture, which leads to limited moderation of surface temperatures and excessive evaporative stress on crops. Due to this direct impact, flash drought studies primarily focus on rapid drought development in the context of agricultural production[2,3,59]. However, research has also recently shown that flash droughts can initiate a sequence of cascading impacts, such as an increased risk for wildfires and heatwave development[15,60]. In light of the results from the global climatology of flash drought occurrence and the rapid land-surface desiccation attributed to rapid drought development, flash drought events have the potential to produce serious impacts beyond agricultural yield loss. Particularly in underdeveloped countries, flash drought that transitions into a long-term drought may lead to an increased risk of famine and destabilization of governments[61,62].

With such a diverse set of meteorological and climatological features having critical roles in the development of flash drought, multiple paths of future studies are needed to understand the drivers of rapid drought intensification across the globe. Furthermore, future research should focus on untangling the complex interactions between flash drought and socioeconomic impacts. Lastly, the results and flash drought events derived from this study provide a reference frame for improvements in flash drought predictability. While examples and discussion of the drivers of flash drought are presented, much future work is needed to advance sub-seasonal predictability of rapid drought development.

## Methods

**Data**. Four global reanalysis data sets were used to generate the spatial and temporal composites of flash drought characteristics. These four data sets include MERRA[26], MERRA-2[27], ERA-Interim[28], and ERA5[29]. The reanalysis data sets were selected based on their global coverage of critical land-surface variables used for flash drought analysis, their temporal focus on the satellite era, and their inclusion of coupled land and atmospheric models.

Daily ET and PET were obtained from each of the four global reanalysis data sets between 1980 and 2015. Daily PET was derived from each of the reanalysis data sets using the Food and Agriculture Organization Penman-Monteith equation[63]. Daily values of the evaporative stress ratio (ESR) were calculated by taking the ratio between daily ET and PET. Mean pentad values of ESR were computed and standardized at each grid point to calculate the standardized ESR (SESR). SESR is given as:

$$\text{SESR}_{ijp} = \frac{\text{ESR}_{ijp} - \overline{\text{ESR}_{ijp}}}{\sigma_{\text{ESR}_{ijp}}} \quad (1)$$

where $\text{SESR}_{ijp}$ (referred to as SESR) is the $z$ score of ESR at a specific grid point $(i, j)$ for a specific pentad $p$, $\overline{\text{ESR}}$ is the mean ESR at a specific grid point $(i, j)$ for a specific pentad $p$ for all years used from the gridded data set, and $\sigma_{\text{ESR}}$ is the standard deviation of ESR at a specific grid point SESR $(i, j)$ for a specific pentad $p$ for all years used from the gridded data set. The temporal change in SESR was also calculated and standardized:

$$(\Delta\text{SESR}_{ijp})_z = \frac{\Delta\text{SESR}_{ijp} - \overline{\Delta\text{SESR}_{ijp}}}{\sigma_{\Delta\text{SESR}_{ijp}}} \quad (2)$$

where $(\Delta SESR_{ijp})_z$ (referred to as $\Delta SESR$) is the z score of the change in SESR from one pentad to another pentad at a specific grid point $(i, j)$ for a specific pentad $p$, $\overline{\Delta SESR}$ is the mean change in SESR values at a specific grid point $(i, j)$ for a specific pentad $p$ for all years used from the gridded data set, and $\sigma_{\Delta SESR}$ is the standard deviation of SESR changes at a specific grid point $(i, j)$ for a specific pentad $p$ for all years used from the gridded data set. SESR and $\Delta SESR$ were both detrended prior to standardizing to account for changes that may have occurred in the drought threshold (SESR) or in individual pentad changes ($\Delta SESR$) over time due to climate change.

**Flash drought identification.** Flash drought events were identified by using a comprehensive identification methodology with a dual emphasis on longevity and impact. Following the guidance for flash drought identification[1], four criteria were used in total, with the first two focusing on the impacts of drought and the last two emphasizing the rapid rate of intensification toward drought[3,9,15,33]. The criteria are:

1) A minimum length of five pentad changes in SESR, equivalent to a length of six pentads (30 days).
2) A final SESR value below the 20th percentile of SESR values.
3a) $\Delta SESR$ must be at or below the 40th percentile between individual pentads.
3b) No more than one $\Delta SESR$ above the 40th percentile following a $\Delta SESR$ that meets criterion 3a.
4) The mean change in SESR during the entire length of the flash drought must be less than the 25th percentile.

Percentiles for criteria 2 and 3 were taken from the distribution of SESR and $\Delta SESR$ at each grid point and specific pentads for all years used from the data set, while percentiles for criterion 4 were taken from the distribution of $\Delta SESR$ at each grid point for pentads that were encompassed within the flash drought event for all years used from the data set.

Criteria 1 and 2 are used to capture land-surface impacts associated with flash drought development. The first criterion (minimum length of 30 days for flash drought) is used to delineate between short-term dry spells and events where rapid drought intensification leads to drought impact. The 20th percentile threshold associated with the second criterion satisfies the drought component of flash drought[1].

Criteria 3 and 4 define thresholds for the rapid rate of intensification toward drought (i.e., the flash component of flash drought). The third criterion is used to identify pentads where conditions are worsening. Criterion 3 is more lenient (40th percentile) than criterion 4 (25th percentile) to account for variability in the rate of drought intensification during flash drought[9]. The more stringent 25th percentile for criterion 4 is applied to the entire duration of the flash drought and is used to ensure that an overall rapid rate of drought intensification occurred during the event.

An example of SESR being used for flash drought identification and its relationship to soil moisture and the development of drought conditions is shown in Supplementary Fig. 9a. Overall, the flash drought methodology shows rapid drought intensification occurring in May and early June with soil moisture depletion during the same time frame. Further, the United States Drought Monitor (USDM) shows a two-category degradation between mid-May and mid-June. In addition to the time series analysis, spatial analysis of the 2012 event shows flash drought development across a large portion of the central United States during the month of May (Supplementary Fig. 9b), similar to the depiction of flash drought development from other studies investigating the 2012 flash drought[2,3].

Regarding the relationship between SESR and soil moisture, the two variables are related via soil water content and ET. As the available soil moisture content is depleted, ET will be reduced at the land surface and SESR will decrease. However, SESR has a greater range of sensitivity for flash drought development compared to soil moisture, as SESR also includes potential evapotranspiration (Supplementary Fig. 9). As such, even when soil moisture becomes largely depleted and ET is significantly reduced, SESR can still decrease due to an increase in PET as land-surface temperatures rise and the vapor pressure deficit increases[15].

With respect to SESR and the USDM, SESR corresponded with deteriorating land-surface conditions with the USDM reaching D1 (moderate drought) shortly after the period of rapid drought intensification and the USDM ultimately reaching D3 (extreme drought) toward the end of summer (Supplementary Fig. 9). Further, SESR began to decline around 2–3 weeks prior to the USDM showing rapid intensification. This result is consistent with several additional case studies comparing SESR and the USDM[9] and the overall lead time that evaporative stress indices provide in flash drought identification[6,7].

**Compositing.** SESR derived from each of the reanalysis data sets was used in the flash drought identification methodology to produce a climatology of flash drought occurrence (Supplementary Figs. 10–13). In the flash drought frequency analysis, a year was registered as a flash drought year if at least one flash drought occurred in a given year, and additional flash droughts in a given year did not contribute to the frequency in Supplementary Figs. 10–13 or in Fig. 1b. After flash drought frequency was determined from each reanalysis, the gridded data sets were composited to combine the results from the four data sets. Because each reanalysis has a different spatial resolution (MERRA: 0.5° × 0.66°, MERRA-2: 0.5° × 0.625°, ERA-

Interim: 0.75° × 0.75°, and ERA5: 0.25° × 0.25°), each flash drought frequency map was bilinearly interpolated to a new grid with a spatial resolution of 0.5° × 0.5°. The mean percentage of flash drought occurrence was then calculated between the four data sets to produce Fig. 1b.

Time series composites were produced by selecting all grid points that were contained within the domains shown in Fig. 1 and averaging the associated variable (Figs. 2 and 3). For the monthly distribution of flash drought occurrence, the starting month for rapid drought intensification was accumulated for all grid points for each month. The percentage of flash droughts that occurred in each month were then averaged between the four reanalyses to produce Fig. 2. For the flash drought spatial coverage time series, all grid points that underwent flash drought for a given year were accumulated. Accumulated flash drought grid points were then converted to a percentage, representing flash drought spatial coverage with respect to the entire domain. This yearly percentage was then averaged between the four reanalyses to produce Fig. 3.

The relative contributions of SPI[35,36] and standardized anomalies of PET toward flash drought development (Fig. 4) were examined by initially finding all flash drought events across grid points contained within the domains shown in Fig. 1 For each individual flash drought event, accumulated precipitation and PET were determined over the time frame of flash drought development. Next, SPI and standardized PET anomalies were calculated over the same time frame as flash drought development using the climatological precipitation and PET data from 1980 to 2015. Once this process was repeated for all four reanalysis data sets, the relative contributions of SPI and standardized PET were partitioned based on thresholds to extract the percentage of large anomalies that occurred during flash drought development (SPI $\leq -1$ and PET $\geq 1$).

Grid points that existed over locations that were too arid or cold were masked on the spatial analysis (Fig. 1b). Arid locations were determined by calculating the aridity index as:

$$AI = \frac{P}{PET} \qquad (3)$$

where $P$ is the average annual precipitation and PET is the average annual potential evapotranspiration from the MERRA-2 data set. Specifically, grid points were masked where the average annual aridity index was below 0.2 (arid and hyper-arid locations) or where the average daily PET was <1 mm per day during the growing season for the Northern Hemisphere (March through October) and Southern Hemisphere (September through April). The aridity threshold was used to place an emphasis on rapid drought development in regions that can transition from more humid to drier environmental conditions and are more likely to experience vegetative, agricultural, or environmental effects from flash drought. In addition, the PET threshold requires regions to have enough evaporative demand throughout the growing season to allow for higher ET rates, sufficient soil moisture depletion, and increased evaporative stress to create rapid drought development.

## Data availability
MODIS land cover data is available at https://lpdaac.usgs.gov. Variables and derived variables used in this study from MERRA and MERRA-2 are available at https://disc.gsfc.nasa.gov, from ERA-Interim are available at https://apps.ecmwf.int/datasets/, and from ERA5 are available at https://cds.climate.copernicus.eu. The global flash drought data generated in this study have been deposited in the Zenodo database at https://doi.org/10.5281/zenodo.5523580.

## Code availability
The code used for this study is available at https://doi.org/10.5281/zenodo.5523698.

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

## Acknowledgements

This work was funded by the NASA Water Resources Program grant 80NSSC19K1266, the National Science Foundation grants OIA-1920946 and OIA-1946093, and the USDA Southern Great Plains Climate Hub.

## Author contributions

J.B.B. conceived of the presented idea. J.I.C. and J.B.B. organized the outline. J.I.C. took the lead in writing the manuscript and provided figures. J.I.C., J.B.B., E.D.H., J.A.O., J.C.F., V.M., X.X., and R.M.R. contributed to the writing of the article.

## Competing interests

The authors declare no competing interests.
