## [Peer Review File · Nature Communications]

Global Distribution, Trends, and Drivers of Flash Drought OccurrenceREVIEWER COMMENTS

Reviewer #1 (Remarks to the Author):

This is an interesting paper that evaluates the spatio-temporal distribution of flash droughts across the globe and their potential drivers. However, due to the unique feature of flash drought (i.e., rapid drought intensification) and the large uncertainty of datasets, it is a great challenge to give a global map of flash droughts. The results presented in this paper are novel, but there still exists substantial uncertainty based on the definition of flash droughts, which makes me hesitate to recommend publication in its present form.

The authors use the evaporative stress ratio (i.e., ET/PET) to identify flash drought. However, there is a lack of physical explanation for the definition of flash droughts, which might induce ample confusion. For example, why use 20th percentile, 40th percentile, and 25th percentile as the thresholds for SESR and delta_SESR? What are the underlying physical meanings for each criteria of flash droughts? How does the soil moisture change during the development of flash droughts? To illustrate these issues, it is better to use an example of flash drought.

Besides, during a severe drought, ET may increase first and then decrease due to the water stress. Under different climatic and hydrological conditions, the changes in ET are quite complex during the development of flash droughts, and it is hard to determine the transmission time. However, it seems that the complex changes of ET are not considered during the identification of flash droughts. I wonder whether the first two criteria can reflect the drought features and thus lead to impacts like the large decrease of crop yields. In addition, ET is also a hydrological variable with large uncertainty in reanalysis datasets. The use of ET rather than other hydrological variables like soil moisture to define flash droughts may induce large certainty. From the global distribution of coefficients of variation for flash droughts, it is clear that the uncertainty of results is too large to identify the hotspots of flash droughts across the globe.

The results for the identification of drivers of flash droughts are also hard to understand. For example, from Fig. 4 we can see around 33% of flash droughts occur with an SPI anomaly, which means around 67% of flash droughts under non-drought conditions. Does it make sense? It seems the author should make more efforts on the definition of flash droughts as well as its uncertainty to give a more convincing global map of flash droughts.

Reviewer #2 (Remarks to the Author):

Summary

This study investigates flash drought characteristics globally using four different reanalysis datasets from 1980-2015. Flash drought occurrence, onset and spatial coverage are evaluated with the standardised evaporative stress ratio (SESR). The presented results show regions with comparatively high flash drought occurrence on all continents and highlight an increase and decrease in flash drought frequency in several regions. Determining the contribution of potential evapotranspiration and precipitation is a major point of discussion in flash drought research and was tackled by the authors by using standardised PET anomalies and the standardised precipitation index. Their findings indicate a similar contribution to flash drought development for both.

The data and methods chosen as well as its well written, organised and comprehensive style contribute to the robustness of this study. This research presents fundamental work towards the understanding of flash drought occurrence globally and extends previous research. Therefore, this study fits the scope of this journal and will be of interest to its readers.

I recommend this work for publication after some minor revisions.

Comments

1: Where contributions of SPI and PET are low (Fig. 9), what could add to flash drought development? Can soil/vegetation specific properties cause decrease in soil moisture, e.g. weak soil water retention and quick percolation? Especially in regard to the low contributions of SPI and PET in SE Australia.

2: Soil moisture is the primarily affected variable during a flash drought which causes impacts on the agriculture. The authors should mention the performance of the SESR compared to soil moisture variations in different climate zones.

3. Regarding the methods, the calculation of the ESR is the standardised ratio of ET to PET. As mentioned by the authors, the calculation is identical with the ESI from Anderson et al. (2007). However, the authors need to explain the different naming of their index, which adds another index to the plethora of indices in drought research.

4. Does soil moisture in the Tropics sufficiently deplete for a flash drought to happen?

5. How were multiple flash drought events in the same year treated when producing Figure 1b)?

6. While the trends in the spatial extent of flash drought in the particular domains provide great insights, the authors should consider adding trends for flash drought frequency. This would be useful, especially in relation with Supplementary Figure 6.

7. Improve readability of figures by increasing font size.

Specific Comments

Figure 4: What does the dashed bold line in each bar chart represent? It looks like the mean of all datasets but is not described in the figure caption.

Reviewer #3 (Remarks to the Author):

This study aims to investigate the spatiotemporal distributions and related mechanisms of global flash drought, a type of drought with rapid onset. While it is a very important and interesting topic, the paper suffers from a few fundamental issues including unclear implication from the defined flash drought, and superficial statements regarding the drivers. It can be a good paper for a professional journal, but it might not be suitable for publication in a Nature journal because of the lack of focus and novelty.

1. The flash drought definition is not new, and it is just an extension from USA to the global area. More importantly, is there any implication for the global distribution of flash drought based on SESR, given that it differs a lot from those results from soil moisture-based flash drought index. A soil moisture-based flash drought index is closely and directly linked with ecological conditions. For SESR, although it is a very complicated procedure for identifying flash drought, its link with ecological or environmental impact of flash drought is not unclear, perhaps partly because it is too complicated. In addition, the SESR method is shown by a HESS paper published earlier this year that it is not able to describe several major flash drought events over USA. I am not totally against SESR, but I just want to mention that obtaining a global picture of flash drought should consider the uncertainty, and illustrate the results with those uncertainties, e.g., when explaining the hotspots, are these really flash drought hotspots? Shall we just call it “evaporative stress flash drought hotspots”?

2. Another major concern is that the driver discussion is not novel. To my opinion, they are good for a professional journal, but not enough for a high rank journal like Nature Communications. The land-

atmosphere coupling, anticyclonic circulation pattern, interannual variability of rainfall, monsoon, ITCZ, ENSO etc have been extensively investigated to explain the mechanism for conventional droughts. What are their unique roles for flash drought? In other word, is there any unique driver for the occurrence of flash drought that is different from conventional drought? If this study can have a novel advancement in this regard, it would be a very insightful paper. Otherwise, it is just listing those well-known facts that are not specifically for flash drought.

3. The relationship between the flash drought trend and the precipitation & PET is too qualitative. Is it possible to distinguish their contributions more objectively and quantitatively? I think a few model simulations are necessary.

4. Why flash droughts increase almost everywhere? More robust analysis is needed.

5. The SESR has been de-trended before identifying flash drought. Does it influence the drought trend analysis?

Reviewer #1 (Remarks to the Author):

The authors use the evaporative stress ratio (i.e., ET/PET) to identify flash drought. However, there is a lack of physical explanation for the definition of flash droughts, which might induce ample confusion. For example, why use 20th percentile, 40th percentile, and 25th percentile as the thresholds for SESR and delta_SESR? What are the underlying physical meanings for each criteria of flash droughts? How does the soil moisture change during the development of flash droughts? To illustrate these issues, it is better to use an example of flash drought.

The authors thank the reviewer for this comment, as a lack of justification was provided in the text for the percentiles thresholds. (Christian et al. 2019) provides a detailed explanation of the percentile selections, but a summary is provided here.

The 20th percentile for SESR at the end of flash drought was required to satisfy the “drought” component of flash drought. This was considered based on (Otkin et al. 2018) and their assessment of flash drought development in which they indicate that any variable that is used to identify flash drought must fall below the 20th percentile for it to be considered drought. In addition, the 20th percentile is widely regarded as the threshold used to represent drought conditions (Svoboda et al. 2002).

The 40th percentile (for pentad-to-pentad changes) and 25th percentile (overall change during flash drought) thresholds were used in tandem to capture the “flash” component of flash drought. The 40th percentile was used to separate periods of worsening conditions (less than 40th percentile) from stable/improving conditions (greater than 40th percentile). The more lenient 40th percentile (compared to the 25th percentile) was also used to account for variability in intensification during flash drought. The 25th percentile was applied to the entire event to ensure that an overall rapid rate of intensification occurred. The 25th percentile was selected after a sensitivity analysis of different thresholds (25th, 20th, 15th, 10th), which ultimately became a categorization of intensification rates during flash drought (Table 1 in Christian et al. 2019).

The following text was added on Lines 483-494:

Criteria 1 and 2 are used to capture land-surface impacts associated with flash drought development. The first criterion (minimum length of 30 days for flash drought) is used to delineate between short-term dry spells and events where rapid drought intensification leads to drought impact. The 20th percentile threshold associated with the second criterion satisfies the “drought” component of flash drought (1).

Criteria 3 and 4 define thresholds for the rapid rate of intensification toward drought (i.e., the “flash” component of flash drought). The third criterion is used to identify pentads where conditions are worsening. Criterion 3 is more lenient (40th percentile) than the criterion 4 (25th percentile) to account for variability in the rate of drought intensification during flash drought (9). The more stringent 25th percentile for criterion 4 is applied to the entire duration of the flash drought and is used to ensure that an overall rapid rate of drought intensification occurred during the event.

The authors also appreciate the suggestion of a flash drought example and have added this to the paper as Supplementary Figure 9. The example provided is from the well-known 2012 flash drought across the central United States using one of the reanalysis datasets as an example (MERRA-2; grid point taken over central Iowa). Soil moisture was added to the time-series to see how soil moisture evolves with SESR. It is shown in the example that the two variables decline simultaneously during flash drought (indicated by the tan shading in the figure). However, SESR has a larger range of standardized values during flash drought as SESR includes evapotranspiration (related to available soil moisture content) and potential evapotranspiration. As such, even when soil moisture is depleted (e.g., between the end of May and early June), SESR can continue to decline due to the additional effects of increasing surface temperature and vapor pressure deficit at the land surface. The information from the United States Drought Monitor (USDM) was also added to the figure to indicate that flash drought identified with SESR not only indicates deteriorating land surface conditions, but also provides a several week lead time over USDM-indicated drought conditions.

The following text was added on Lines 495-513:

An example of SESR being used for flash drought identification and its relationship to soil moisture and the development of drought conditions is shown in Supplementary Figure 9. Overall, the flash drought methodology shows rapid drought intensification occurring in May and early June with soil moisture depletion during the same timeframe. Further, the United States Drought Monitor (USDM) shows a two-category degradation between mid-May and mid-June.

Regarding the relationship between SESR and soil moisture, the two variables are related via soil water content and ET. As the available soil moisture content is depleted, ET will be reduced at the land surface and SESR will decrease. However, SESR has a greater range of sensitivity for flash drought development compared to soil moisture, as SESR also includes potential evapotranspiration (Supplementary Figure 9). As such, even when soil moisture becomes largely depleted and ET is significantly reduced, SESR can still decrease due to an increase in PET as land surface temperatures rise and the vapor pressure deficit increases (15).

With respect to SESR and the USDM, SESR corresponded with deteriorating land surface conditions with the USDM reaching D1 (moderate drought) shortly after the period of rapid drought intensification and the USDM ultimately reaching D3 (extreme drought) toward the end of summer (Supplementary Figure 9). Further, SESR began to decline around 2-3 weeks prior to the USDM showing rapid intensification. This result is consistent with several additional case studies comparing SESR and the USDM (9) and the overall lead time that evaporative stress indices provide in flash drought identification (6, 7).

Supplementary Figure 9. SESR and 0-100 cm standardized soil moisture from MERRA-2, as well as the USDM drought category in central Iowa, United States during 2012. The tan color indicates the time period of flash drought.

Besides, during a severe drought, ET may increase first and then decrease due to the water stress. Under different climatic and hydrological conditions, the changes in ET are quite complex during the development of flash droughts, and it is hard to determine the transmission time. However, it seems that the complex changes of ET are not considered during the identification of flash droughts. I wonder whether the first two criteria can reflect the drought features and thus lead to impacts like the large decrease of crop yields. In addition, ET is also a hydrological variable with large uncertainty in reanalysis datasets. The use of ET rather than other hydrological variables like soil moisture to define flash droughts may induce large certainty. From the global distribution of coefficients of variation for flash droughts, it is clear that the uncertainty of results is too large to identify the hotspots of flash droughts across the globe.

The authors thank the reviewer for bringing up this important point, as this response of ET is a critical reason why SESR was used for flash drought analysis. ET alone would be difficult to use to determine rapid drought development due to its complex evolution during drought development. However, SESR normalizes ET with potential evapotranspiration (PET) to account for these complex changes. To illustrate this relationship, an example is used from the 2010 southwestern Russia flash drought (Figs. R1 and R2; Christian et al. 2020). As the reviewer stated, ET first begins to rise at the beginning of flash drought due to increased PET. However, because SESR is the ratio of ET and PET, SESR declines in early June regardless of the enhanced ET as PET increases to values that are well above average. As such, the complex changes of ET are accounted for during flash drought development by including PET in the identification process.

Figure R1. Domain-averaged standardized anomalies of the evaporative stress ratio (SESR), 2-meter air temperature, 10-meter wind speed, and 2-meter water vapor pressure deficit for 2010 from MERRA-2. The tan color indicates the time period of rapid drought development and the orange color indicates the time period of heatwave conditions (figure taken from Christian et al. 2020).

Figure R2. Domain-averaged standardized anomalies of EVI from MODIS and ET and PET from MERRA-2 for 2010. The tan color indicates the time period of rapid drought development and the orange color indicates the time period of heatwave conditions (figure taken from Christian et al. 2020).

Regarding the first two flash drought criteria (rapid intensification lasting at least 30 days and drought severity being at or below the 20th percentile threshold), the flash drought methodology has been shown to represent impacts at the land surface by comparing SESR derived flash drought to the USDM in numerous case studies (Christian et al. 2019) and extensive case study analysis (the 2012 central United

States flash drought - Basara et al. 2019, and the 2010 southwestern Russia flash drought - Christian et al. 2020). Specifically for the 2010 flash drought case, SESR and the criteria were shown to strongly correspond with land surface desiccation as shown via a vegetation index from satellite observations.

While the manuscript discusses SESR and the methodology's utility as an early drought indicator and its ability to correspond with impacts via the USDM, the following text was added on Line 93 for an additional description/reference of the methodology's link to land surface impact:
and land surface desiccation via satellite observations (15).

With respect to uncertainty of flash drought frequency, flash drought hotspots actually had very little uncertainty/variability between reanalysis datasets (e.g., the Sahel, Great Rift Valley, India, and Northern Australia) with coefficient of variation values below 0.3. **This was a core purpose of this study - to identify flash drought hotspots in robust manner.** However, highlighting additional local hotspot regions with higher variability among reanalysis datasets (e.g., central United States, western Russia, northeastern China) does not nullify the results, but rather highlights the complexity of these regions in regard to flash drought development. Given that other hydrologic variables, such as soil moisture, are heavily modeled in reanalysis datasets (especially root zone soil moisture), some regions with high uncertainty/variability between reanalysis datasets will also exist if using a variable such as soil moisture. Similar concerns were also suggested by additional reviewers, and Supplementary Figure 4 was added to the manuscript to highlight the strong relationship between SESR and soil moisture.

The following text was added on Lines 237-249:

While SESR and the flash drought methodology used in this study have consistently shown to identify flash drought (3, 9, 15, 33) for several notable events across different regions, it is critical to evaluate its performance in capturing flash drought with respect to land surface impact. A key hydrological variable used to determine vegetative impact during flash drought analysis is soil moisture (14, 19). While SESR indirectly includes soil water content via the magnitude of ET, examining soil moisture directly provides insight into the magnitude of land surface desiccation from SESR-derived flash drought events. After calculating the average soil moisture percentile at the end of all flash drought events between the four reanalysis datasets, it was found that soil moisture was depleted to the 20th percentile or lower in 11 of the 15 study regions and depleted to the 25th percentile or lower in some portion of all 15 study regions (Supplementary Figure 4). Overall, the analysis indicates that flash droughts identified via evaporative stress consistently capture rapid drought intensification that leads toward depleted soil moisture content.

Supplementary Figure 4. Mean soil moisture percentile at the end of flash droughts between 1980 and 2015 from the four reanalysis datasets. The dotted black line represents the contour for the 20th percentile.

The results for the identification of drivers of flash droughts are also hard to understand. For example, from Fig. 4 we can see around 33% of flash droughts occur with an SPI anomaly, which means around 67% of flash droughts under non-drought conditions. Does it make sense? It seems the author should make more efforts on the definition of flash droughts as well as its uncertainty to give a more convincing global map of flash droughts.

The authors thank the reviewer for this comment as there was a miscommunication in the paper. The results in Fig. 4 only show when large anomalies in SPI (or PET) exist during flash drought, not just any below-average anomaly. Overall, the results indicate the **nearly equal importance of highly anomalous PET in the development of flash drought** - the key point being made. However, the importance of PET in flash drought development was not well illustrated in Fig. 4 or discussed in the text. To emphasize the importance of PET, a new figure was added that highlights the frequency of the lead driver during flash drought (SPI anomaly or PET anomaly; Supplementary Figure 3). This figure better communicates the relative importance of large, anomalous PET during flash drought development in addition to a lack of rainfall. In addition, text was modified between Lines 197-207 to better communicate the role of *large* anomalies on flash drought development.

Text was also added on Lines 250-258 to discuss the importance of PET during flash drought.

Flash drought development is driven by the simultaneous occurrence of precipitation deficits and above-average evaporative demand (1). The results in Fig. 4 and Supplementary Fig. 3 show strong evidence for the nearly equal importance of highly anomalous PET and a lack of precipitation during flash drought. Enhanced PET was the lead driver during flash drought development (47.8% of the time) nearly as often as negative precipitation anomalies (52.2% of the time; Supplementary Figure 3). As

such, while large precipitation deficits remain a key driver in flash drought development, excessive evaporative demand at the land surface can also strongly contribute toward rapid drought intensification when only small to moderate precipitation deficits occur during the flash drought development period.

Supplementary Figure 3. Percentage of flash drought events with SPI or PET as the lead driver during flash drought development from the four reanalysis datasets (different colored bars) for each of the domains outlined in black on the map. The black dotted lines represent the mean between all four reanalysis datasets.

- Basara, J. B., J. I. Christian, R. A. Wakefield, J. A. Otkin, E. H. Hunt, and D. P. Brown, 2019: The evolution, propagation, and spread of flash drought in the Central United States during 2012. *Environ Res Lett*, **14**, 084025, <https://doi.org/10.1088/1748-9326/ab2cc0>.
- Christian, J. I., J. B. Basara, J. A. Otkin, E. D. Hunt, R. A. Wakefield, P. X. Flanagan, and X. Xiao, 2019: A Methodology for Flash Drought Identification: Application of Flash Drought Frequency Across the United States A Methodology for Flash Drought Identification: Application of Flash Drought Frequency Across the United States. *J Hydrometeorol*, **20**, 833–846, <https://doi.org/10.1175/jhm-d-18-0198.1>.
- Christian, J. I., J. B. Basara, E. D. Hunt, J. A. Otkin, and X. Xiao, 2020: Flash drought development and cascading impacts associated with the 2010 Russian heatwave. *Environ Res Lett*, **15**, 094078, <https://doi.org/10.1088/1748-9326/ab9faf>.
- Otkin, J. A., M. Svoboda, E. D. Hunt, T. W. Ford, M. C. Anderson, C. Hain, and J. B. Basara, 2018: Flash Droughts: A Review and Assessment of the Challenges Imposed by Rapid Onset Droughts in the United States. *B Am Meteorol Soc*, **99**, 911–919, <https://doi.org/10.1175/bams-d-17-0149.1>.

Svoboda, M., and Coauthors, 2002: The Drought Monitor. *B Am Meteorol Soc*, **83**, 1181–1190, <https://doi.org/10.1175/1520-0477-83.8.1181>.

Reviewer #2 (Remarks to the Author):

1: Where contributions of SPI and PET are low (Fig. 4), what could add to flash drought development? Can soil/vegetation specific properties cause decrease in soil moisture, e.g. weak soil water retention and quick percolation? Especially in regard to the low contributions of SPI and PET in SE Australia.

The authors thank the reviewer for this comment. The authors agree that soil type and land cover type may increase the complexity of flash drought development traditionally associated with large precipitation deficits with anomalously high PET. The following text was added to discuss possible complications toward flash drought development in this region on Lines 259-267:

In regions where the contributions of large SPI and PET anomalies are low, soil type and land cover type may increase the complexity of flash drought development (9). Of particular note are the study regions over southeastern Australia, the Iberian Peninsula, and Asia Minor which had the lowest contributions from SPI and PET anomalies during flash drought. These three study regions are located in semi-arid environments that either directly border arid/hyper-arid environments or barren regions (Fig. 1a). As such, sparse vegetation, soils with weak water retention, and advection from arid or hyper-arid regions may complicate flash drought development in these regions with respect to rapid drought intensification primarily associated with large precipitation deficits and above-average evaporative demand.

2: Soil moisture is the primarily affected variable during a flash drought which causes impacts on the agriculture. The authors should mention the performance of the SESR compared to soil moisture variations in different climate zones.

The authors thank and appreciate the reviewer for this comment, and accordingly have added a new global analysis illustrating the average soil moisture percentile at the end of flash drought (Supplementary Figure 4). The results indicate that flash droughts identified by SESR contain low soil moisture values at the end of rapid intensification period.

The following text was added on Lines 237-249:

While SESR and the flash drought methodology used in this study have consistently shown to identify flash drought (3, 9, 15, 33) for several notable events across different regions, it is critical to evaluate its performance in capturing flash drought with respect to land surface impact. A key hydrological variable used to determine vegetative impact during flash drought analysis is soil moisture (14, 19). While SESR indirectly includes soil water content via the magnitude of ET, examining soil moisture directly provides insight into the magnitude of land surface desiccation from SESR-derived flash drought events. After calculating the average soil moisture percentile at the end of all flash drought events between the four reanalysis datasets, it was found that soil moisture was depleted to the 20th percentile or lower in 11 of the 15 study regions and depleted to the 25th percentile or lower in some portion of all 15 study regions (Supplementary Figure 4). Overall, the analysis indicates that flash

droughts identified via evaporative stress consistently capture rapid drought intensification that leads toward depleted soil moisture content.

Supplementary Figure 4. Mean soil moisture percentile at the end of flash droughts between 1980 and 2015 from the four reanalysis datasets. The dotted black line represents the contour for the 20th percentile.

In addition, an example time-series of SESR and soil moisture was added to the paper to illustrate the relationship between declining SESR and soil moisture depletion (Supplementary Figure 9).

Supplementary Figure 9. SESR and 0-100 cm standardized soil moisture from MERRA-2, as well as the USDM drought category in central Iowa, United States during 2012. The tan color indicates the time period of flash drought.

The following text was added on Lines 495-506:

An example of SESR being used for flash drought identification and its relationship to soil moisture and the development of drought conditions is shown in Supplementary Figure 9. Overall, the flash drought methodology shows rapid drought intensification occurring in May and early June with soil moisture depletion during the same timeframe. Further, the United States Drought Monitor (USDM) shows a two-category degradation between mid-May and mid-June.

Regarding the relationship between SESR and soil moisture, the two variables are related via soil water content and ET. As the available soil moisture content is depleted, ET will be reduced at the land surface and SESR will decrease. However, SESR has a greater range of sensitivity for flash drought development compared to soil moisture, as SESR also includes potential evapotranspiration (Supplementary Figure 9). As such, even when soil moisture becomes largely depleted and ET is significantly reduced, SESR can still decrease due to an increase in PET as land surface temperatures rise and the vapor pressure deficit increases (15).

3. Regarding the methods, the calculation of the ESR is the standardised ratio of ET to PET. As mentioned by the authors, the calculation is identical with the ESI from Anderson et al. (2007). However, the authors need to explain the different naming of their index, which adds another index to the plethora of indices in drought research.

Text was added to clarify the similarities and differences of ESI and SESR on Lines 84-87:

SESR is similar to the evaporative stress index (ESI; 30, 31) in which both indices are calculated by the ratio of ET and PET and then standardized. However, SESR is primarily derived using reanalysis-based variables while the ESI is derived using satellite observations.

4. Does soil moisture in the Tropics sufficiently deplete for a flash drought to happen?

From the newly added soil moisture figure (Supplementary Figure 4), the answer is a resounding yes. At the end of flash drought, soil moisture drops to the 20th percentile or lower across the tropics.

5. How were multiple flash drought events in the same year treated when producing Figure 1b)?

The authors thank the reviewer for this comment. Clarification text was added on Lines 517-519:

In the flash drought frequency analysis, a year was registered as a flash drought year if at least one flash drought occurred in a given year, and additional flash droughts in a given year did not contribute to the frequency in Supplementary Figures 10-13 or in Figure 1b.

6. While the trends in the spatial extent of flash drought in the particular domains provide great insights, the authors should consider adding trends for flash drought frequency. This would be useful, especially in relation with Supplementary Figure 6.

The authors thank the reviewer for this important comment. The original goal of this study was to perform an analysis of flash drought frequency. However, such an analysis is incredibly challenging on a grid-by-grid point basis due to the low frequency of flash drought. For example, at a specific grid point, flash drought years can be represented as 1's (flash drought in the given year) or 0's (no flash drought in the given year). However, a trend line could not be added to a time series of 1's and 0's over a 36 year time period. Ultimately, due to this issue, trends in spatial extent of flash drought were calculated to approximately represent changing flash drought frequency with time.

7. Improve readability of figures by increasing font size.

Font size has been increased for Figures 1-4, and Supplementary Figures 7 and 8.

Figure 4: What does the dashed bold line in each bar chart represent? It looks like the mean of all datasets but is not described in the figure caption.

The authors thank the reviewer for catching this omission - new text was added in the caption for Figure 4 to clarify that the bold line is the mean of all datasets.

The black dotted lines represent the mean between all four reanalysis datasets.

Reviewer #3 (Remarks to the Author):

1. The flash drought definition is not new, and it is just an extension from USA to the global area. More importantly, is there any implication for the global distribution of flash drought based on SESR, given that it differs a lot from those results from soil moisture-based flash drought index. A soil moisture-based flash drought index is closely and directly linked with ecological conditions. For SESR, although it is a very complicated procedure for identifying flash drought, its link with ecological or environmental impact of flash drought is not unclear, perhaps partly because it is too complicated. In addition, the SESR method is shown by a HESS paper published earlier this year that it is not able to describe several major flash drought events over USA. I am not totally against SESR, but I just want to mention that obtaining a global picture of flash drought should consider the uncertainty, and illustrate the results with those uncertainties, e.g., when explaining the hotspots, are these really flash drought hotspots? Shall we just call it "evaporative stress flash drought hotspots"?

The authors would like to thank the reviewer for bringing up several important points regarding SESR and the analysis. Each point is addressed in detail below.

The flash drought definition is not new, and it is just an extension from USA to the global area.

The authors agree this flash drought definition is not new, but it is for this reason that it was used for a global analysis. We would like to clarify that the aim of the current work is not to propose or develop a new method rather to examine the global hot spots of flash droughts. While most flash drought definitions have only been used once or twice in the scientific literature, there has already been considerable efforts in developing, refining, and exploring the utility of the SESR method in preparation for a global-scale analysis of flash drought (Christian et al. 2019a, Christian et al. 2019b, Basara et al. 2019, Christian et al. 2020, Osman et al. 2021).

More importantly, is there any implication for the global distribution of flash drought based on SESR, given that it differs a lot from those results from soil moisture-based flash drought index. A soil moisture-based flash drought index is closely and directly linked with ecological conditions. For SESR, although it is a very complicated procedure for identifying flash drought, its link with ecological or environmental impact of flash drought is not unclear, perhaps partly because it is too complicated.

The authors thank the reviewer for this important comment. To illustrate the relationship between SESR and soil moisture, an example time-series was added to the paper (Supplementary Figure 9). SESR has a similar response to soil moisture due to its inclusion of ET, but the addition of PET (evaporative demand) in the ratio of SESR increases the sensitivity of SESR to flash drought development. The USDM is also shown in Supplementary Figure 9 to illustrate the ecological response foreshadowed by the rapid decline of SESR. Several other case studies are shown in Christian et al. 2019a highlighting SESR's ability to represent ecological degradation and drought conditions as represented by the USDM for several different regions and years across the United States. Further, Christian et al. 2020 show that rapid declines in SESR correspond with rapid land surface desiccation represented by a vegetation index from satellite observations.

The following text was added on Lines 495-513:

An example of SESR being used for flash drought identification and its relationship to soil moisture and the development of drought conditions is shown in Supplementary Figure 9. Overall, the flash drought methodology shows rapid drought intensification occurring in May and early June with soil moisture depletion during the same timeframe. Further, the United States Drought Monitor (USDM) shows a two-category degradation between mid-May and mid-June.

Regarding the relationship between SESR and soil moisture, the two variables are related via soil water content and ET. As the available soil moisture content is depleted, ET will be reduced at the land surface and SESR will decrease. However, SESR has a greater range of sensitivity for flash drought development compared to soil moisture, as SESR also includes potential evapotranspiration (Supplementary Figure 9). As such, even when soil moisture becomes largely depleted and ET is significantly reduced, SESR can still decrease due to an increase in PET as land surface temperatures rise and the vapor pressure deficit increases (15).

With respect to SESR and the USDM, SESR corresponded with deteriorating land surface conditions with the USDM reaching D1 (moderate drought) shortly after the period of rapid drought intensification and the USDM ultimately reaching D3 (extreme drought) toward the end of summer (Supplementary Figure 9). Further, SESR began to decline around 2-3 weeks prior to the USDM showing rapid intensification. This result is consistent with several additional case studies comparing SESR and the USDM (9) and the overall lead time that evaporative stress indices provide in flash drought identification (6, 7).

Supplementary Figure 9. SESR and 0-100 cm standardized soil moisture from MERRA-2, as well as the USDM drought category in central Iowa, United States during 2012. The tan color indicates the time period of flash drought.

In addition, Supplementary Figure 4 was added to the paper to show soil moisture conditions at the end of flash droughts identified in the four reanalysis datasets. The results here indicate that flash droughts identified with SESR are characterized by a large depletion of soil moisture.

The following text was added on Lines 237-249:

While SESR and the flash drought methodology used in this study have consistently shown to identify flash drought (3, 9, 15, 33) for several notable events across different regions, it is critical to evaluate its performance in capturing flash drought with respect to land surface impact. A key hydrological variable used to determine vegetative impact during flash drought analysis is soil moisture (14, 19). While SESR indirectly includes soil water content via the magnitude of ET, examining soil moisture directly provides insight into the magnitude of land surface desiccation from SESR-derived flash drought events. After calculating the average soil moisture percentile at the end of all flash drought events between the four reanalysis datasets, it was found that soil moisture was depleted to the 20th percentile or lower in 11 of the 15 study regions and depleted to the 25th percentile or lower in some portion of all 15 study regions (Supplementary Figure 4). Overall, the analysis indicates that flash droughts identified via evaporative stress consistently capture rapid drought intensification that leads toward depleted soil moisture content.

Supplementary Figure 4. Mean soil moisture percentile at the end of flash droughts between 1980 and 2015 from the four reanalysis datasets. The dotted black line represents the contour for the 20th percentile.

In addition, the SESR method is shown by a HESS paper published earlier this year that it is not able to describe several major flash drought events over USA.

The authors thank the reviewer for this comment; however, we respectfully disagree with this suggestion. Given that a couple of authors on this study were also coauthors on the HESS paper, the following comments are based on intimate knowledge with the development of the Osman et al. (2021) study.

First, Figure 6 in Osman et al. (2021) shows SESR was able to broadly capture flash drought for 1988, 2011, and 2012, with Montana in 2017 being the lone exception. Further, capturing “impact” from identified flash drought is critical for a flash drought methodology (Otkin et al. 2018). While the USDM is not designed to capture flash drought (Osman et al. 2021), it can be used here as a reference for drought impact. In Figure 6, the SESR method is the only method of the 5 other indices to capture flash drought as a subset of drought (Otkin et al. 2018) in which SESR identified flash drought as a subset of USDM identified flash drought. All other indices often show flash drought development across more than 50% of the CONUS land area during the four years analyzed in Osman et al. (2021) in areas that did not even experience drought conditions via the USDM.

In addition, the SESR method has already been shown to consistently identify flash drought via case studies (the 2012 central US flash drought - Basara et al. 2019; the 2010 western Russia flash drought - Christian et al. 2020) and time series of SESR during multiple flash drought events in Christian et al. 2019a (including the 2011 flash drought). No time-series are shown for any of the 5 other indices investigated in Osman et al. (2021; except for a lone time-series of SMVI for one case study), so their ability for flash drought identification remains relatively unknown.

The SESR method has been shown to sufficiently capture flash drought across several different regions and years, and is suitable for a global scale analysis of flash drought.

I am not totally against SESR, but I just want to mention that obtaining a global picture of flash drought should consider the uncertainty, and illustrate the results with those uncertainties, e.g., when explaining the hotspots, are these really flash drought hotspots? Shall we just call it “evaporative stress flash drought hotspots”?

The authors thank the reviewer for this comment. The importance of communicating uncertainty is the very reason why four global reanalyses were used in this study instead of one reanalysis dataset. The variability indicated in Fig. S1 highlights this uncertainty of flash drought frequency across the globe. Several key flash drought hotspot regions had very little uncertainty/variability between reanalysis datasets (e.g., the Sahel, Great Rift Valley, India, and Northern Australia) with coefficient of variation values below 0.3. However, other local hotspot regions with higher variability between reanalysis datasets (e.g., central United States, western Russia, northeastern China) highlights the complexity of these regions in regard to flash drought development.

2. Another major concern is that the driver discussion is not novel. To my opinion, they are good for a professional journal, but not enough for a high rank journal like Nature Communications. The land-atmosphere coupling, anticyclonic circulation pattern, interannual variability of rainfall, monsoon, ITCZ, ENSO etc have been extensively investigated to explain the mechanism for conventional droughts. What are their unique roles for flash drought? In other word, is there any unique driver for the occurrence of flash drought that is different from conventional drought? If this study can have a novel advancement in this regard, it would be a very insightful paper. Otherwise, it is just listing those well-known facts that are not specifically for flash drought.

The authors absolutely agree that understanding the drivers of flash droughts is a critical issue. However, this topic warrants, at a minimum, an entire study and possibly several individual studies for each driver. Further, it is essential to first know when and where flash drought events occur in order to subsequently identify the primary drivers of these events. This is an important step in that process.

The authors also agree that some of the drivers of drought development are similar to their role in flash drought development. However, while drought development usually revolves around a lack of precipitation, the discussion section focuses on how subseasonal and climatic features specifically contribute toward flash drought development in both a lack of rainfall and increased PET.

To clarify how this discussion focuses on drivers specifically with flash drought development, the following text was added on Lines 397-405:

Many of the meteorological drivers and climatic features previously discussed (land-atmosphere coupling, anticyclones, interannual variability of rainfall, monsoons, the ITCZ, and ENSO) can also contribute toward conventional drought development (i.e., drought development on seasonal timescales or longer; 56-58). However, while drought is primarily characterized by a lack of precipitation, flash drought development occurs due to a combination of below-average precipitation and enhanced evaporative demand (1). As such, the unique contribution of these features toward flash drought

development involves not only the suppression of rainfall, but the additional influence of above-average evaporative demand to rapidly deplete moisture and lead to rapid land surface desiccation.

While the discussion focuses on possible drivers of flash drought via scientific literature or supplemental figures (e.g., Figs. S3, S4, and S6) in the context of precipitation deficits and above-average evaporative demand, this critical topic is also in our future plans of research (Lines 430-432).

3. The relationship between the flash drought trend and the precipitation & PET is too qualitative. Is it possible to distinguish their contributions more objectively and quantitatively? I think a few model simulations are necessary.

The authors thank the reviewer for this comment. Because the trend analyses were on annual timescales, it would be challenging to synthesize in-depth quantitative analysis that would be meaningful and concise enough to include in the manuscript. However, to further evaluate the contribution of precipitation and PET toward flash drought development from another perspective, Supplementary Figure 3 was added to the manuscript. Supplementary Figure 3 highlights the frequency of the lead driver during flash drought (precipitation deficit or anomalously high PET).

The authors also agree that model simulations are a critical topic to explore with flash drought but would necessitate an entire study to effectively address the relationship between flash drought trends and the associated sensitivity to precipitation and PET over time.

Supplementary Figure 3. Percentage of flash drought events with SPI or PET as the lead driver during flash drought development from the four reanalysis datasets (different colored bars) for each of the domains outlined in black on the map. The black dotted lines represent the mean between all four reanalysis datasets.

4. Why flash droughts increase almost everywhere? More robust analysis is needed.

Flash drought spatial coverage had statistically significant increases over time for 6 of the 15 regions, and statistically significant decreases for 3 of the 15 regions (Fig. 3).

The original goal of this study was to perform an analysis of flash drought frequency. However, such an analysis is incredibly challenging on a grid-by-grid point basis due to the low frequency of flash drought. For example, at a specific grid point, flash drought years can be represented as 1's (flash drought in the given year) or 0's (no flash drought in the given year). However, a trend line could not be added to a time series of 1's and 0's over a 36 year time period. Ultimately, due to this issue, trends in spatial extent of flash drought were calculated to approximately represent changing flash drought frequency with time.

5. The SESR has been de-trended before identifying flash drought. Does it influence the drought trend analysis?

The authors thank the reviewer for this comment. The trend analysis was redone with data that was not detrended (Fig. R3), and the trends are nearly identical with the p-values only changing slightly for the 15 study regions (compare Fig. 3 with Fig. R3). Because detrending the data protects artificially enhanced results in the trend analysis from external factors (e.g., increasing temperatures with time that influence PET, increasing precipitation variability with time that influences ET), the authors have kept the detrended data and results in the manuscript.

Figure 3. Mean flash drought spatial coverage (percent) from the four reanalysis datasets (black line) for each of the domains outlined in black on the map. The green shaded area represents the variability (standard deviation) between the four reanalyses and the thicker blue line represents the trend line for flash drought spatial coverage. P-values highlighted in red are statistically significant trends at the 90% confidence level using the Mann-Kendall test.

Figure R3. Same as Figure 3, but using SESR and Δ SESR that is not detrended.

- Basara, J. B., J. I. Christian, R. A. Wakefield, J. A. Otkin, E. H. Hunt, and D. P. Brown, 2019: The evolution, propagation, and spread of flash drought in the Central United States during 2012. *Environ Res Lett*, **14**, 084025, <https://doi.org/10.1088/1748-9326/ab2cc0>.
- Christian, J. I., J. B. Basara, J. A. Otkin, E. D. Hunt, R. A. Wakefield, P. X. Flanagan, and X. Xiao, 2019a: A Methodology for Flash Drought Identification: Application of Flash Drought Frequency Across the United States. *J Hydrometeorol*, **20**, 833–846, <https://doi.org/10.1175/jhm-d-18-0198.1>.
- Christian, J. I., J. B. Basara, J. A. Otkin, and E. D. Hunt, 2019b: Regional characteristics of flash droughts across the United States. *Environ Res Commun*, **1**, 125004, <https://doi.org/10.1088/2515-7620/ab50ca>.
- Christian, J. I., J. B. Basara, E. D. Hunt, J. A. Otkin, and X. Xiao, 2020: Flash drought development and cascading impacts associated with the 2010 Russian heatwave. *Environ Res Lett*, **15**, 094078, <https://doi.org/10.1088/1748-9326/ab9faf>.
- Osman, M., B. F. Zaitchik, H. S. Badr, J. I. Christian, T. Tadesse, J. A. Otkin, and M. C. Anderson, 2021: Flash drought onset over the contiguous United States: sensitivity of inventories and trends to quantitative definitions. *Hydrol Earth Syst Sc*, **25**, 565–581, <https://doi.org/10.5194/hess-25-565-2021>.
- Otkin, J. A., M. Svoboda, E. D. Hunt, T. W. Ford, M. C. Anderson, C. Hain, and J. B. Basara, 2018: Flash Droughts: A Review and Assessment of the Challenges Imposed by Rapid Onset Droughts in the United States. *B Am Meteorol Soc*, **99**, 911–919, <https://doi.org/10.1175/bams-d-17-0149.1>.

REVIEWERS' COMMENTS

Reviewer #1 (Remarks to the Author):

Review of “Global distribution, trends, and drivers of flash drought occurrence” by Christian et al.

The authors have put forth an admirable effort to revise this paper. However, my biggest concern is still the novelty of this work which has not been fully addressed. Either the definition or global analysis of flash droughts have been discussed by published studies (e.g., Christian et al. 2019 or Koster et al. 2019). In addition, the analysis of drivers of flash drought development is also too simple and does not contribute much to understanding the physical processes of flash drought at global or continental scale (see detailed comments below). Therefore, I am not convinced that the novelty of this study provides the type of step change in understanding that one would expect for work published in this journal.

I appreciate that the authors evaluate the robustness of global pattern of flash drought based on four reanalysis datasets. However, it is known that ET is the one of the most uncertain hydrological variables in reanalysis datasets. The calculation of PET is also very complicated and large uncertainties would be further involved in calculation of flash drought. That is why the coefficients of variation for flash drought in most regions are too large to provide a convincing global map of flash drought. In other words, this also implies that the flash droughts based on four reanalysis datasets are not consistent with each other over most regions across the world. I was wondering whether two or three hot spots with small coefficients of variation can convince readers of the estimation of global pattern or trend of flash drought, which is one the most important conclusions of this work. This is another major limitation of study.

Another major concern is the lack of observations. I was also wondering why the authors have not include more observations in their assessment? An inclusion of observational-based estimates would make the message of this paper much stronger and robust. I appreciate that these observations are not as widely available, but at least some comparisons for key variables are needed so that the reader can trust the estimation of the global pattern and trend of flash drought.

The part regarding drivers of flash drought is rather weak. As the authors mentioned that “flash drought development occurs due to a combination of below-average precipitation and enhanced evaporative demand”, which is rational and easy to understand. However, I doubt the motivation of analysis the unique contribution of precipitation deficits and evaporative demand anomalies and the conclusion that the contribution of above-average evaporative demand is similar to that of precipitation. How could the high evaporative demand alone drive flash drought and there are no obvious precipitation deficits during the development of flash drought? Are there any historical flash drought cases or published

studies supporting such kind of physical mechanism? If so, how about the impact on crop yields in such cases? There are also large number of flash drought either experience precipitation deficits and evaporative demand anomalies, which is also lack of evidence of real world cases and hard to understand. I was also wondering the impacts of such kind of flash drought on agricultural production.

Reviewer #2 (Remarks to the Author):

The authors have addressed all my comments from the first review and I am satisfied with their responses and additional work they put in to make this a sound paper. I have no further comments to be made and recommend the paper in its current form for publication.

Reviewer #3 (Remarks to the Author):

As I mentioned in previous round of review, this manuscript does not have enough novelty and broad impact for publishing in Nature Communications. The revision clarified a few technical issues, but the scientific contribution is still limited. The flash drought identification method is not new. The global distribution and trend analysis are straightforward and simple, but they have large uncertainties. The driver discussions are the same as normal droughts, thus do not provide any new implications. Therefore, I have to recommend for rejection.

1. Global distribution has a large uncertainty. Most hotspots in Figure 1 have a large variations across different reanalysis data (Fig. S1). Moreover, the global distribution based on SESR differs a lot from that based on soil moisture. As other reviewer pointed out, the soil moisture-based flash drought identification has clearer implication for agriculture and ecology. Then, how to understand global distribution in Figure 1? Does it really represent global hot spots that we should focus?

2. Given the large uncertainty in global distribution of the occurrence, the trend analysis is also questionable. The trends also conflict with a few regional studies from the published literature. The uncertainty analysis should be emphasized. Even if we believe the calculation of the trends is ok, what are the implications for showing these regional trends? Do they mean they will continue these trends in the future? Droughts usually result from internal variability, and a 36-year (1980-2015) period might not be enough to exclude the internal variability. In other words, we do not know whether these upward or downward trends are just natural oscillations given the short data records. The record length is

especially important for flash droughts because they usually have longer return periods than normal droughts.

3. Another major concern is the discussion of the drivers of flash drought. I can hardly believe this manuscript provide any breakthrough. Listing potential drivers for different regions could be a good technical report, but not enough for a scientific paper, especially for a paper in Nature Communications. The readers should know something they do not know from a Nature Communications paper. Unfortunately, I cannot find those messages in this paper. I re-emphasize a few questions that are not answered: The land-atmosphere coupling, anticyclonic circulation pattern, interannual variability of rainfall, monsoon, ITCZ, ENSO etc have been extensively investigated to explain the mechanism for conventional droughts. What are their unique roles for flash drought? In other word, is there any unique driver for the occurrence of flash drought that is different from normal drought? The role of ET (as simply explained in the revision) is also not new, while it is mentioned in almost all published flash drought papers.

4. Droughts usually have several characteristic measures, e.g., frequency, duration, severity. To fully understand global distribution, these characteristics should be presented and discussed. This will also verify whether the definition is appropriate for flash drought.

5. Physical explanation of SESR is still not clear. The spatial plot (not a point time series) for an extreme flash drought event (e.g., 2012 central US flash drought) is necessary for verifying the proposed index.

6. "However, SESR has a greater range of sensitivity for flash drought development compared to soil moisture, as SESR also includes potential evapotranspiration (Supplementary Figure 9). As such, even when soil moisture becomes largely depleted and ET is significantly reduced, SESR can still decrease due to an increase in PET as land surface temperatures rise and the vapor pressure deficit increases (15)." Why does the range matter? Soil moisture can also reflect the influence of ET or PET. If the soil moisture drops to a very low value, it becomes as a normal drought instead of flash drought.

Reviewer #1:

The authors have put forth an admirable effort to revise this paper. However, my biggest concern is still the novelty of this work which has not been fully addressed. Either the definition or global analysis of flash droughts have been discussed by published studies (e.g., Christian et al. 2019 or Koster et al. 2019). In addition, the analysis of drivers of flash drought development is also too simple and does not contribute much to understanding the physical processes of flash drought at global or continental scale (see detailed comments below). Therefore, I am not convinced that the novelty of this study provides the type of step change in understanding that one would expect for work published in this journal.

The authors thank the reviewer for providing feedback on the first revision that motivated the inclusion of the previously added analyses to the paper. As discussed with reviewer #3 in the first revision, we absolutely agree that a deeper understanding of the drivers of flash drought is critically important. However, it is essential to first know when and where flash drought events occur in order to subsequently identify the primary drivers of these events. Identifying global flash drought hotspots is an important step in that process.

I appreciate that the authors evaluate the robustness of global pattern of flash drought based on four reanalysis datasets. However, it is known that ET is the one of the most uncertain hydrological variables in reanalysis datasets. The calculation of PET is also very complicated and large uncertainties would be further involved in calculation of flash drought. That is why the coefficients of variation for flash drought in most regions are too large to provide a convincing global map of flash drought. In other words, this also implies that the flash droughts based on four reanalysis datasets are not consistent with each other over most regions across the world. I was wondering whether two or three hot spots with small coefficients of variation can convince readers of the estimation of global pattern or trend of flash drought, which is one the most important conclusions of this work. This is another major limitation of study.

The authors thank the reviewer for this comment about uncertainty. As previously stated in the first revision, the most prominent flash drought hotspots (the Sahel, Great Rift Valley, India, and Northern Australia) had minimal uncertainty/variability between reanalysis datasets, and moderate uncertainty for additional local hotspots (e.g., central United States, western Russia, northeastern China) highlights the complexity of flash drought development in these regions. Determining flash drought from other variables (such as root zone soil moisture) are also highly modeled, and will also likely produce uncertainty between different datasets. However, given that uncertainty will exist regardless of the variable used, text was added to discuss the uncertainty and the role it will have on the results.

L416-426: The climatology of flash droughts provided in this study are derived from evaporative stress. While evaporative stress is related to other hydrologic variables used for flash drought analysis (e.g., soil moisture), it is important to note that the results of this study may differ from those that use a different variable or flash drought identification methodology. However, key hotspots shown in this study align with a previous study using soil moisture and a different identification methodology for the Northern Hemisphere (37), indicating the consistency of major flash drought hotspots regardless of the variable or methodology used. Local hotspots regions that vary between evaporative stress driven flash drought and soil moisture driven flash drought suggest a greater complexity of flash drought development in these regions. As such, the

results and conclusions in this study should be primarily limited to evaporative stress based flash drought events.

Another major concern is the lack of observations. I was also wondering why the authors have not include more observations in their assessment? An inclusion of observational-based estimates would make the message of this paper much stronger and robust. I appreciate that these observations are not as widely available, but at least some comparisons for key variables are needed so that the reader can trust the estimation of the global pattern and trend of flash drought. The authors thank the reviewer for their comment on observations and flash drought. In terms of direct observational comparison (e.g., reanalyses ET vs. observational ET), observations of ET are quite limited spatially and temporally. Further and more importantly, the representativeness of in situ observations is highly limited in spatial extent due to variability in land cover type. As such, microscale/local comparisons of reanalyses ET with observation ET will have little meaning/importance for a global-scale analysis.

On the other hand, comparisons of evaporative stress to other related variables (precipitation, temperature, etc.) have already been explored in previous studies. These comparisons have been made in SESR-based papers (Christian et al. 2019, Christian et al. 2020) and ESI-based papers (Anderson et al. 2015, Nguyen et al. 2019, Otkin et al. 2013). While evaporative stress is directly affected by many variables, such studies have shown that evaporative stress has a relationship with key flash drought variables for various years and study regions.

Lastly, it's important to note that reanalysis datasets do assimilate/incorporate a wide range of observational datasets for key variables in flash drought development (e.g., precipitation and temperature). While ET is heavily modeled in reanalysis datasets, it still relies on fundamental land surface data (e.g., precipitation, vapor pressure deficit, etc.). As such, reanalysis datasets provide the best approach (currently) for global-scale flash drought analysis that spans a period long enough for a statistical climatology.

Anderson, M. C. et al. Comparison of satellite-derived LAI and precipitation anomalies over Brazil with a thermal infrared-based Evaporative Stress Index for 2003–2013, *Journal of Hydrology* 526, 287-302 (2015).

Christian, J. I. et al. A methodology for flash drought identification: application of flash drought frequency across the United States. *Journal of Hydrometeorology* 20, 833–846 (2019).

Christian, J. I., Basara, J. B., Hunt, E. D., Otkin, J. A. & Xiao, X. Flash drought development and cascading impacts associated with the 2010 Russian heatwave. *Environ. Res. Lett.* 15, 094078 (2020).

Nguyen, H. et al. Using the evaporative stress index to monitor flash drought in Australia. *Environmental Research Letters* 14, 064016 (2019).

Otkin, J. A. et al. Examining Rapid Onset Drought Development Using the Thermal Infrared–Based Evaporative Stress Index. *Journal of Hydrometeorology* 14, 1057–1074 (2013).

The part regarding drivers of flash drought is rather weak. As the authors mentioned that “flash drought development occurs due to a combination of below-average precipitation and enhanced evaporative demand”, which is rational and easy to understand. However, I doubt the motivation of analysis the unique contribution of precipitation deficits and evaporative demand anomalies

and the conclusion that the contribution of above-average evaporative demand is similar to that of precipitation. How could the high evaporative demand alone drive flash drought and there are no obvious precipitation deficits during the development of flash drought? Are there any historical flash drought cases or published studies supporting such kind of physical mechanism? If so, how about the impact on crop yields in such cases? There are also large number of flash drought either experience precipitation deficits and evaporative demand anomalies, which is also lack of evidence of real world cases and hard to understand. I was also wondering the impacts of such kind of flash drought on agricultural production.

The authors thank the reviewer for this comment regarding the contribution of ET and PET during flash drought development. This was also noted in the previous revision, and text was added during this previous revision to clarify this miscommunication in the paper. Figure 4 only shows when strong anomalies of precipitation deficits or high PET occur during flash drought, and does not indicate that negative anomalies of precipitation are not present during flash drought. For example, in Figure 4 for the United States, 42% of flash droughts had a significant precipitation anomaly ($SPI \leq -1$) during flash drought. Further, 83% of flash droughts in the US had a precipitation deficit (e.g., $SPI < 0$), and the last 17% of flash droughts had SPI between 0 and 0.5 (near-normal precipitation) with anomalously high PET.

Currently, the coauthors are working on specific analyses that show it is possible to have flash drought development with near-normal precipitation if anomalously high vegetative green-up occurs. In such a scenario, the increased biomass requires increased soil moisture to maintain vegetation health; even near-normal precipitation amounts are not sufficient to prevent a rapid drying of soil moisture and flash drought development, specifically when coupled with high PET. Overall, this type of condition can occur whereby flash drought develops with near-normal precipitation when high PET is present through enhanced desiccation of the soil column but is relatively rare by comparison. Even so, it remains possible within overlapping environmental situations. Future work beyond this study will need to explore these events to explore all the mechanisms that may lead to flash drought in the situation of high PET and near-normal precipitation.

Reviewer #3:

1. Global distribution has a large uncertainty. Most hotspots in Figure 1 have a large variations across different reanalysis data (Fig. S1). Moreover, the global distribution based on SESR differs a lot from that based on soil moisture. As other reviewer pointed out, the soil moisture-based flash drought identification has clearer implication for agriculture and ecology. Then, how to understand global distribution in Figure 1? Does it really represent global hot spots that we should focus?

The authors thank the reviewer for this comment on uncertainty. A similar comment was provided from reviewer #1 about uncertainty, and our response is shared here as well. As previously stated in the first revision, the most prominent flash drought hotspots (the Sahel, Great Rift Valley, India, and Northern Australia) had very little uncertainty/variability between reanalysis datasets, and moderate uncertainty for additional local hotspots (e.g., central United States, western Russia, northeastern China) highlights the complexity of flash drought development in these regions. Determining flash drought from other variables (such as root zone soil moisture) are also highly modeled, and will also likely produce uncertainty between different

datasets. However, given that uncertainty will exist regardless of the variable used, text was added to discuss the uncertainty and the role it will have on the results.

L416-426: The climatology of flash droughts provided in this study are derived from evaporative stress. While evaporative stress is related to other hydrologic variables used for flash drought analysis (e.g., soil moisture), it is important to note that the results of this study may differ from those that use a different variable or flash drought identification methodology. However, key hotspots shown in this study align with a previous study using soil moisture and a different identification methodology for the Northern Hemisphere (37), indicating the consistency of major flash drought hotspots regardless of the variable or methodology used. Local hotspots regions that vary between evaporative stress driven flash drought and soil moisture driven flash drought suggest a greater complexity of flash drought development in these regions. As such, the results and conclusions in this study should be primarily limited to evaporative stress based flash drought events.

2. Given the large uncertainty in global distribution of the occurrence, the trend analysis is also questionable. The trends also conflict with a few regional studies from the published literature. The uncertainty analysis should be emphasized. Even if we believe the calculation of the trends is ok, what are the implications for showing these regional trends? Do they mean they will continue these trends in the future? Droughts usually result from internal variability, and a 36-year (1980-2015) period might not be enough to exclude the internal variability. In other words, we do not know whether these upward or downward trends are just natural oscillations given the short data records. The record length is especially important for flash droughts because they usually have longer return periods than normal droughts.

The authors thank the reviewer for this comment about the implication of the trend analysis. We agree that the trend results and implications should be restricted to the 36-year period and should be careful to avoid suggesting these trends will continue. We also agree that datasets with extended data records may reveal that these trends could be embedded within natural oscillations over longer time periods (e.g., 100 years). As such, the following text was added in the trend analysis results:

L192-196: It is important to note that the results of the trend analysis only apply to the 36-year period used in the study (1980-2015) and do not indicate that these trends will extend into the future. Further, notable trends revealed for the analysis may also be embedded within internal variability of the climate due to the relatively short study period and may change with a longer period of record.

3. Another major concern is the discussion of the drivers of flash drought. I can hardly believe this manuscript provide any breakthrough. Listing potential drivers for different regions could be a good technical report, but not enough for a scientific paper, especially for a paper in Nature Communications. The readers should know something they do not know from a Nature Communications paper. Unfortunately, I cannot find those messages in this paper. I re-emphasize a few questions that are not answered: The land-atmosphere coupling, anticyclonic circulation pattern, interannual variability of rainfall, monsoon, ITCZ, ENSO etc have been extensively investigated to explain the mechanism for conventional droughts. What are their unique roles for flash drought? In other word, is there any unique driver for the occurrence of

flash drought that is different from normal drought? The role of ET (as simply explained in the revision) is also not new, while it is mentioned in almost all published flash drought papers. The authors thank the reviewer for providing their concern regarding the novelty for Nature Communications. The discussion of drivers was not the core purpose of the paper, but provided a broader connection between the primary results of this study (which were the global distribution of flash drought, trends of flash drought, and initial key variables that drive flash drought (precipitation deficit vs. evaporative demand)) and atmospheric/oceanic drivers of flash drought.

As stated in the previous revision, conventional drought discussion via the above mentioned drivers primarily revolve around a lack of precipitation. Flash drought requires the unique combination between precipitation deficits and above-average evaporative demand, which is the focus of the discussion in this paper.

Newly added text on L407-415 was provided on the first revision to clarify this issue.

4. Droughts usually have several characteristic measures, e.g., frequency, duration, severity. To fully understand global distribution, these characteristics should be presented and discussed. This will also verify whether the definition is appropriate for flash drought.

The authors thank the reviewer for the comments on drought characteristics. We agree that other characteristic measures can be quantified for flash drought (duration and severity). However, flash droughts differ from drought in the sense that drought focuses on magnitude and flash drought focuses on rate of change. Unlike drought that may continue for a long, unrestricted period of time, duration of flash drought is predefined both from a perspective of the identification methodology and the general definition of flash drought (Otkin et al. 2018). The minimum length of flash drought required is 30 days in this study, with most rapid intensification events lasting no longer than 2 months. As such, the average duration of flash drought across the globe resides between 30 and 60 days, and such information does not provide critical improvements in the understanding of the global distribution of flash drought.

Regarding severity, the coauthor group has introduced a simple method for quantifying severity in Christian et al. 2019. However, severity categories have not been well studied, and many severity calculations exist in the literature. Until further advances are made in establishing a consistent technique/method for flash drought severity/intensity, results of climatological flash drought severity across the globe will be very limited. We believe future work beyond this study should be dedicated in this research area to further our understanding in quantifying flash drought severity/intensity, similar to recently published work in Otkin et al. (2021).

Christian, J. I. et al. A methodology for flash drought identification: application of flash drought frequency across the United States. *Journal of Hydrometeorology* 20, 833–846 (2019).

Otkin, J. A. et al. Flash droughts: a review and assessment of the challenges imposed by rapid-onset droughts in the United States. *Bulletin of the American Meteorological Society* 99, 911–919 (2018).

Otkin, J.A. et al. Development of a Flash Drought Intensity Index. *Atmosphere* 12, 741 (2021).

5. Physical explanation of SESR is still not clear. The spatial plot (not a point time series) for an extreme flash drought event (e.g., 2012 central US flash drought) is necessary for verifying the proposed index.

The authors thank the reviewer for these comments.

A physical explanation of SESR is added on L84-89:

SESR represents the overall evaporative stress on the environment. SESR becomes positive when ample soil moisture is available, surface temperatures and vapor pressure deficit are lower, and cloudy skies are present (reduced shortwave radiation). In contrast, SESR becomes negative when soil moisture is depleted, surface temperatures and vapor pressure deficit increase, and clear skies are present (increased shortwave radiation).

The spatial plot of the 2012 flash drought was added as panel b) on Supplementary Figure 9 and associated text was added in L520-523.

Supplementary Figure 9. The time series (a) shows *SESR* and 0-100 cm standardized soil moisture from MERRA-2, as well as the USDM drought category in southeastern Iowa, United States during 2012. The tan color indicates the time period of flash drought in a). The spatial plot (b) shows the month in which flash drought began. The black outline in b) shows the location of the grid point used in a).

6. “However, SESR has a greater range of sensitivity for flash drought development compared to soil moisture, as SESR also includes potential evapotranspiration (Supplementary Figure 9). As such, even when soil moisture becomes largely depleted and ET is significantly reduced, SESR can still decrease due to an increase in PET as land surface temperatures rise and the vapor pressure deficit increases (15).” Why does the range matter? Soil moisture can also reflect the influence of ET or PET. If the soil moisture drops to a very low value, it becomes as a normal drought instead of flash drought.

The authors want to first clarify that soil moisture is an incredibly useful variable in flash drought development. That being said, evaporative stress based metrics can have some advantages specifically for flash drought detection due to the inclusion of PET. Soil moisture can be decoupled from the atmosphere such that soil moisture can remain constant for a period of time even if PET increases. Therefore, the increased sensitivity of SESR to land surface and atmospheric response allows a slightly increased response time to flash drought development. In addition, evaporative stress provides a composited approach to monitor land surface impact due to the simultaneous occurrence of rapid soil moisture depletion and increased atmospheric demand. Because of this, evaporative stress metrics have been a fundamental approach to identify flash drought (Anderson et al. 2013, Basara et al. 2019, Christian et al. 2019, Christian et al. 2020, Nguyen et al. 2019, Otkin et al. 2013, Otkin et al. 2014, Otkin et al. 2016).

- Anderson, M. C. et al. An Intercomparison of Drought Indicators Based on Thermal Remote Sensing and NLDAS-2 Simulations with U.S. Drought Monitor Classifications. *Journal of Hydrometeorology* 14, 1035–1056 (2013).
- Basara, J. B. et al. The evolution, propagation, and spread of flash drought in the Central United States during 2012. *Environmental Research Letters* 14, 084025 (2019).
- Christian, J. I. et al. A methodology for flash drought identification: application of flash drought frequency across the United States. *Journal of Hydrometeorology* 20, 833–846 (2019).
- Christian, J. I., Basara, J. B., Hunt, E. D., Otkin, J. A. & Xiao, X. Flash drought development and cascading impacts associated with the 2010 Russian heatwave. *Environ. Res. Lett.* 15, 094078 (2020).
- Nguyen, H. et al. Using the evaporative stress index to monitor flash drought in Australia. *Environmental Research Letters* 14, 064016 (2019).
- Otkin, J. A. et al. Examining Rapid Onset Drought Development Using the Thermal Infrared–Based Evaporative Stress Index. *Journal of Hydrometeorology* 14, 1057–1074 (2013).
- Otkin, J. A., Anderson, M. C., Hain, C. & Svoboda, M. Examining the Relationship between Drought Development and Rapid Changes in the Evaporative Stress Index. *Journal of Hydrometeorology* 15, 938–956 (2014).
- Otkin, J. A. et al. Assessing the evolution of soil moisture and vegetation conditions during the 2012 United States flash drought. *Agricultural and Forest Meteorology* 218–219, 230–242 (2016).